# Still Competitive: Revisiting Recurrent Models for Irregular Time Series Prediction

**Ankitkumar Joshi**                                                                 *ahjoshi@pitt.edu*
*Department of Computer Science*
*University of Pittsburgh*

**Milos Hauskrecht**                                                                 *milos@pitt.edu*
*Department of Computer Science*
*University of Pittsburgh*

**Reviewed on OpenReview:** *https: // openreview. net/ forum? id= YLoZA77QzR*

## Abstract

Modeling irregularly sampled multivariate time series is a persistent challenge in domains like healthcare and sensor networks. While recent works have explored a variety of complex learning architectures to solve the prediction problems for irregularly sampled time series, it remains unclear what the true benefits of some of these architectures are, and whether clever modifications of simpler and more efficient RNN-based algorithms are still competitive, i.e. they are on par with or even superior to these methods. In this work, we propose and study GRUwE: Gated Recurrent Unit with Exponential basis functions, that builds upon RNN-based architectures for observations made at irregular times. GRUwE supports both regression-based and event-based predictions in continuous time. GRUwE works by maintaining a Markov state representation of the time series that updates with the arrival of irregular observations. The Markov state update relies on two reset mechanisms: (i) observation-triggered reset to account for the new observation, and (ii) time-triggered reset that relies on learnable exponential decays, to support the predictions in continuous time. Our empirical evaluations across several real-world benchmarks on next-observation and next-event prediction tasks demonstrate that GRUwE can indeed achieve competitive or superior performance compared to the recent state-of-the-art (SOTA) methods. Thanks to its simplicity, GRUwE offers compelling advantages: it is easy to implement, requires minimal hyper-parameter tuning efforts, and significantly reduces the computational overhead in the online deployment.

## 1 Introduction

Multivariate time series and their models are central to understanding and analyzing real-world dynamical systems. While traditional sequence models typically assume regularly spaced observations, real-world data such as healthcare records or sensor measurements, are often irregularly sampled. In such settings, events or measurements may occur at uneven time intervals and under varying contextual conditions (e.g., specialized laboratory tests in clinical environments may be performed only during acute episodes). The key challenge is to develop time-series models that can accurately capture dependencies among multiple variables and events occurring at irregular times. In this work, we study this problem in the context of both time-series forecasting and event prediction.

A variety of models ranging from classic statistical and modern deep learning frameworks have been developed to support time-series prediction task for irregular settings. Early approaches replaced the irregularly observed data with regularly spaced observations by inferring their values at the regular time points. Classic statistical auto-regressive models (AR, ARMA, ARIMA) Shumway & Stoffer (2017) or latent space models, such as, Linear Dynamical Systems (LDS) Kalman (1963) models could then be applied to support prediction at future regular times. To support prediction at arbitrary future prediction times, various smoothing

and interpolation methods Liu & Hauskrecht (2015) were developed. More recently, classic time-series models have been gradually replaced with various types of modern neural architectures capable of handling irregularly sampled observations Siami-Namini et al. (2018). The existing methods include extensions of *RNN approaches* Che et al. (2018); Mei & Eisner (2017), and continue with *differential equation approaches* Rubanova et al. (2019); De Brouwer et al. (2019); Chen et al. (2018); Schirmer et al. (2022); Becker et al. (2019), *attention-based approaches* Shukla & Marlin (2021); Chen et al. (2023), *graph-based approaches* Zhang et al. (2022; 2024); Yalavarthi et al. (2024) and *state-space modeling approaches* Smith et al. (2023); Gu & Dao (2023).

Recent work on irregular multivariate time series has largely gravitated toward increasingly complex architectures such as continuous-time Neural ODEs Rubanova et al. (2019); Kidger et al. (2020); Chen et al. (2023), transformers Zuo et al. (2020); Zhang et al. (2020); Shukla & Marlin (2021); Yang et al. (2022), and graph-based models Zhang et al. (2024); Yalavarthi et al. (2024); Li et al. (2025); Luo et al. (2025). While these approaches introduce powerful modeling mechanisms, they often require intricate training pipelines, extensive hyperparameter tuning, and incur substantial computational overhead. Moreover, inductive biases folded in the complex architectures may be difficult to interpret or analyze. These factors may limit the benefits of such architectures in many realistic settings particularly, when training data are limited or deployment environments are resource constrained (e.g., bedside monitoring systems or embedded clinical devices), where the efficiency of inference is critical for their deployment. This raises a natural question: *could carefully designed, time-series prediction models that are rooted in simpler architectures, still achieve comparable or even superior predictive performance at significantly lower computational cost?* We investigate this question by proposing **GRUwE** (Gated Recurrent Unit with Exponential basis functions), pronounced as "groovy", a relatively simple yet effective GRU-based architecture that maintains the Markov state representation of the multivariate process that is updated at irregular observation times. GRUwE models the effect of time with the help of learnable exponential basis functions. State updates are affected by both: (i) the arrival of new observations, and (ii) the elapsed time since the previous observation. This dual mechanism allows the model to naturally accommodate influence of past observations and irregular time intervals between them. To make predictions at an arbitrary future time, GRUwE reuses the same exponential basis functions that are used for state updates.

As noted above, GRUwE supports both time series forecasting and event prediction tasks. For event prediction, we define a Temporal Point Process (TPP) model with the help of a Conditional Intensity Function (CIF), which characterizes the instantaneous rate of event occurrence. Following prior works by Du et al. (2016); Mei & Eisner (2017), the CIF is modeled by applying a Softplus transformation to the output of a neural network regression model. This formulation allows GRUwE's predictions at arbitrary future times to directly define the event intensity. We compare the GRUwE-based event prediction model against state-of-the-art CIF-based approaches, including methods based on RNN models Du et al. (2016); Mei & Eisner (2017) and attention-based models Zhang et al. (2020); Zuo et al. (2020); Yang et al. (2022).

The main contributions of this work are:

- We propose GRUwE, an irregular time series model, that maintains a compact Markov state representation, defines two reset state update mechanisms to capture dependencies inherent in multivariate time series and supports continuous-time inference.

- We perform a comprehensive evaluation of GRUwE against state-of-the-art baselines on next-observation and next-event prediction tasks across multiple real-world datasets. Our results show that GRUwE performs competitively on both tasks, achieves new state-of-the-art performance on two datasets for next-observation prediction, and attains the highest overall rank for next-event prediction.

- We demonstrate that GRUwE's compact Markov state representation leads to substantial improvements in computational efficiency during online deployment.

## 2 Related Work

In this section, we review existing work on time series prediction for irregularly sampled multivariate time series data and contrast them to the proposed GRUwE model. Our review is structured along key architectural designs that have been proposed for time-series and event prediction tasks.

**Recurrent neural network (RNN) approaches** exploit efficient sequential hidden-state updates for prediction, with the hidden state serving as a compact, Markovian summary of all past observations. Classical RNN models assume regularly sampled time series and therefore require interpolation schemes to handle irregularly sampled data. Several extensions adapt recurrent models to continuous time by updating the hidden state only at observation times, including Recurrent Marked Temporal Point Processes (RMTPP) Du et al. (2016), GRU-D Che et al. (2018), FullyNN Omi et al. (2019), and the Neural Hawkes Process (NHP) Mei & Eisner (2017). RMTPP extends classic RNN architecture by adding a parametric conditional intensity function that predicts the distribution of the next event time directly from the RNN's hidden state. NHP, based on LSTM Hochreiter & Schmidhuber (1997) architecture, uses exponential decays to a predicted target cell state to derive a continuous-time cell state representation, which is eventually used to represent the intensity function to model time-varying intensities and complex temporal point-process dynamics. GRU-D extends GRU by explicitly modeling missingness and irregular sampling using learnable decay mechanisms that impute both inputs and hidden states based on the time since each variable was last observed. Missing observations are inferred in GRU-D using decay mechanisms that revert to a target value, typically determined by the mean value of the variable. The GRU-D and GRUwE models are similar in their use of the exponential decay functions for modeling the effect of elapsed time on the hidden state. However, GRU-D uses decay functions to infer missing observation values. As a result, it may propagate estimation errors for missing observations in time. In contrast, GRUwE does not interpolate missing observations and uses exponential decay functions to directly model latent variable trajectories both for the state update and prediction module.

**Differential equation approaches** offer another widely used framework for modeling multivariate time series in continuous time. These methods define latent dynamics with the help of differential equations. Neural ODEs Chen et al. (2018) model temporal evolution using ordinary differential equations parameterized by neural networks. Extensions such as Latent ODE and ODE-RNN Rubanova et al. (2019) combine these dynamics with variational and recurrent architectures to handle irregular observations. A key limitation of Neural ODEs is that their solutions depend on fixed initial conditions that cannot easily adapt to observed data. Neural controlled differential equations Kidger et al. (2020); De Brouwer et al. (2019) address this limitation by allowing continuous modulation based on the input observations. However, most differential equation–based approaches require external numerical solvers, significantly increasing training and inference costs. Schirmer et al. (2022); Becker et al. (2019) avoid the use of numerical solvers for continuous-time dynamics by modeling latent state transitions using linear stochastic differential equations that admit closed-form solutions. In contrast, GRUwE captures latent dynamics through learnable exponential decay functions, requiring no numerical solvers and imposing no linearity assumptions on the underlying state evolution.

**Temporal attention** methods eliminate the need to approximate a latent Markov state representation and instead learn prediction functions directly from the observed inputs. Models such as mTAND Shukla & Marlin (2021), THP Zuo et al. (2020), SAHP Zhang et al. (2020), A-NHP Yang et al. (2022) embeds time into fixed-size learnable vectors and learn relationship between irregular observations and prediction timepoints in continuous time. ContiFormer Chen et al. (2023) combines Neural ODEs with the continuous-time attention mechanism to model irregular observations.

**Graph Neural Networks (GNN) approaches** focus on modeling interactions and dependencies among individual time series. T-PatchGNN Zhang et al. (2024) uses a patch-based mechanism on univariate time series to enhance local feature extraction and use graph neural network to learn relationships among different time series. GraFITi Yalavarthi et al. (2024) first converts irregular samples to a special bipartite graph structure and cast the prediction problem as an edge weight prediction. HyperIMTSLi et al. (2025) improves on the GraFITi method by replacing its fixed bipartite graph with a learnable, heterogeneous hypergraph that jointly models variable–time relations, enables richer message passing, and produces more expressive irregular time series representations. Other graph-based temporal Zhang et al. (2022), spatiotemporal Marisca et al. (2022) and diffusion-based Tashiro et al. (2021) models have been proposed to address irregularly spaced observations. Overall, the main difference between GRUwE and existing temporal attention and GNN approaches is that GRUwE maintains a Markov state representation that enables efficient update and deployment in real-time settings supporting sequential prediction tasks. Typically the above transformer and graph-based models require one to buffer sequences of past observations to encode a newly

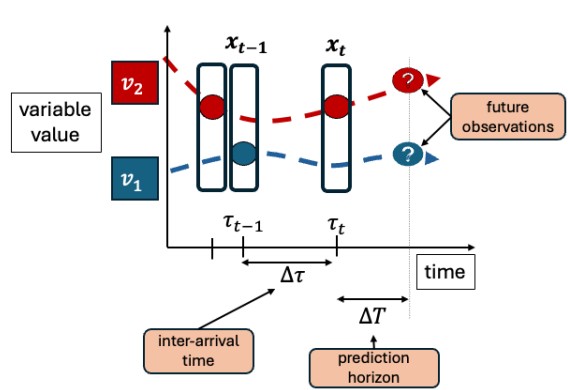

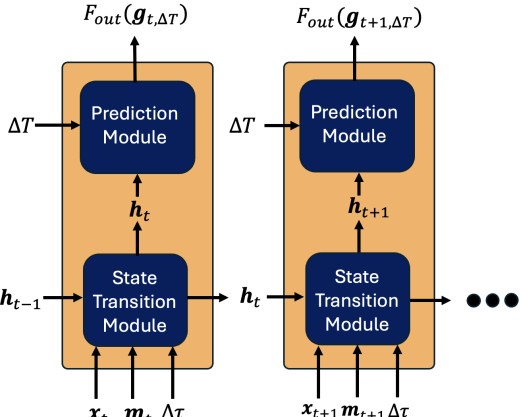

(a) Abstract diagram of the next-observation prediction problem for irregular time series. The true continuous dynamics function of the variables $\{v_1, v_2\}$ (dotted line) are unknown; the model only sees their values at irregular time points to formulate future predictions.

(b) GRUwE's overall architecture. At each observation time point, $\{\mathbf{x}_t, \mathbf{m}_t, \Delta\tau\}$ are processed to update the previous hidden state $(\mathbf{h}_{t-1} \rightarrow \mathbf{h}_t)$. The prediction module make a prediction using the hidden state $\mathbf{h}_t$ and prediction horizon $\Delta T$.

Figure 1: Description of the problem and overview of the model architecture.

arrived observation and reprocess all the past observations for every inference step from scratch which may become a computational bottleneck when sequences of past observation become very long. In contrast, GRUwE updates its state using only the previous state and the newly arrived observation, making it simple, efficient, and practical for real-time deployment.

## 3 Problem Setting

Our goal is to model $D$-dimensional multivariate time series with irregularly spaced observations. For clarity, consider the case $D = 2$ with variables $v_1$ and $v_2$, whose latent temporal dynamics are illustrated as dotted trajectories in Figure 1a. The underlying generative process governing these variables is unknown. Observations occur at non-uniform time points (shown as dots), and may not be synchronized across variables; for instance, $v_1$ may be missing when $v_2$ is observed. At each time step $t \in \mathbb{Z}$ (corresponding to continuous time $\tau_t \in \mathbb{R}$), we define an input vector $\mathbf{x}_t \in \mathbb{R}^D$ where observed entries contain values and unobserved entries are masked. The time elapsed between successive observation times, also known as inter-arrival time, is denoted by $\Delta\tau = (\tau_t - \tau_{t-1}) \in \mathbb{R}_+$. To support prediction at any future time, we define a prediction horizon $\Delta T \in \mathbb{R}_+$, representing how far ahead (relative to $\tau_t$) the prediction should be made. In the *next observation prediction* task, the model generates a predicted observation vector $\hat{\mathbf{x}}_{t, \Delta T}$ at time $\tau_t + \Delta T$, conditioned on the history of all observations up to time step $t$. In the *next event prediction* setting, the TPP model outputs event intensities $\lambda(\tau_t + \Delta T)$, given the event history up to time step $t$.

## 4 GRUwE: Gated Recurrent Unit with Exponential Basis Functions

Our goal is to define and learn a prediction model $f(\mathbf{x}_{1:t}, \boldsymbol{\tau}_{1:t}, \Delta T)$ that takes a history of a $D$-dimensional multivariate time series $\mathbf{x}_{1:t}$ made at times $\boldsymbol{\tau}_{1:t}$ respectively, and predicts future values of time series at time $\tau_t + \Delta T$.

**Markov State.** Since the observation history grows over time, we approximate it using a fixed-size state representation $\mathcal{H}_t$ that summarizes what is known about the process until time step $t$. If $\mathcal{H}_t$ accurately summarizes the history of observations, the process becomes Markovian, rendering past observations conditionally independent of the future given the current state. This property allows the state to be updated using only the previous state and the newly arrived observation, which greatly simplifies real-time deployment. The state $\mathcal{H}_t$ of our GRUwE model consists of a latent vector $\mathbf{h}_t$ representing dependencies among different

time series variables and their values. Thus, $\mathcal{H}_t = \{\mathbf{h}_t\}$ at time $\tau_t$ summarizes all the observations made until time $t$.

**State Transition Module.** As new observations arrive in time, the state $\mathcal{H}_t$ representing the information seen so far must be updated. The transition between the two consecutive GRUwE's states is implemented using two complementary update mechanisms. The first accounts for the time elapsed since the previous observation. We refer to it as to the time-triggered reset mechanism. The second mechanism incorporates the influence of a newly observed input. We refer to it as to the observation-triggered reset mechanism. In the following, we describe the design of these mechanisms in more detail.

**Time-Triggered Reset Mechanism** models the effect of elapsed time between two consecutive observations using learnable exponential decay functions. The decay functions take the previous hidden state and the time difference between previous and new observations to update the state. The decay functions are defined as:

$$\boldsymbol{\gamma}(\boldsymbol{\Delta\tau}) = \exp\{-\max(\mathbf{0}, \mathbf{W}_\gamma \boldsymbol{\Delta\tau} + \mathbf{b}_\gamma)\},$$

where $\boldsymbol{\Delta\tau} = (\tau_t - \tau_{t-1}) \cdot \mathbb{1}$ denotes the elapsed time vector, with $\mathbb{1}$ representing a vector of ones that broadcasts the scalar time difference across all state dimensions. The learnable parameters $\mathbf{W}_\gamma$ and $\mathbf{b}_\gamma$ control the rate at which each component of the hidden state decays as a function of elapsed time. The time-decayed hidden state is then computed as:

$$\mathbf{g}_{t-1,\Delta\tau} = \boldsymbol{\gamma}(\boldsymbol{\Delta\tau}) \odot \mathbf{h}_{t-1}, \tag{1}$$

where $\odot$ denotes element-wise multiplication operation.

**Observation-Triggered Reset Mechanism** models the influence of a new observation on the state. The input (new observation) at each time-step $t$ is defined by an input vector $\mathbf{x}_t$ and a mask vector $\mathbf{m}_t$ indicating if an individual variable $i$ is observed at time-step $t$:

$$m_{t,i} = \begin{cases} 1, & \text{if } x_{t,i} \text{ is observed,} \tag{2} \\ 0, & \text{otherwise.} \tag{3} \end{cases}$$

Since observations for multivariate times series may arrive at different times, the values missing in the specific input vector are masked. That is, at each time-step, the recurrent unit takes two vectors as the input: 1) a mask vector ($\mathbf{m}_t$) and 2) input vector ($\mathbf{x}_t$) where missing values are substituted with zeros:

$$\mathbf{x}'_t = \mathbf{m}_t \odot \mathbf{x}_t . \tag{4}$$

Computing the hidden state ($\mathbf{h}_t$) at $\tau_t$, given $\mathbf{g}_t$, $\mathbf{m}_t$ and $\mathbf{x}_t$ involves a set of updates similar to the ones found in the standard GRU unit:

$$\begin{aligned} \mathbf{z}_t &= \sigma(\mathbf{W}_z \mathbf{x}'_t + \mathbf{U}_z \mathbf{g}_{t-1,\Delta\tau} + \mathbf{V}_z \mathbf{m}_t + \mathbf{b}_z), \\ \mathbf{r}_t &= \sigma(\mathbf{W}_r \mathbf{x}'_t + \mathbf{U}_r \mathbf{g}_{t-1,\Delta\tau} + \mathbf{V}_r \mathbf{m}_t + \mathbf{b}_r), \\ \tilde{\mathbf{h}}_t &= \tanh(\mathbf{W}_h \mathbf{x}'_t + \mathbf{U}_h(\mathbf{r}_t \odot \mathbf{g}_{t-1,\Delta\tau}) + \mathbf{V}_h \mathbf{m}_t + \mathbf{b}), \\ \mathbf{h}_t &= (1 - \mathbf{z}_t) \odot \mathbf{g}_{t-1,\Delta\tau} + \mathbf{z}_t \odot \tilde{\mathbf{h}}_t. \end{aligned}$$

Note that $\mathbf{W}$s, $\mathbf{U}$s, $\mathbf{V}$s and $\mathbf{b}$s are learnable parameters of the model. The parameters let us fit the hidden state component and observations so that the dependencies among time series most important for the prediction are captured.

**Prediction module.** Our main goal is to predict future observations in continuous time given the past observation sequence. At each time step $t$, we rely only on the information encoded in the model's state $\mathcal{H}_t$ to support prediction. To enable flexible prediction across arbitrary future time points, we introduce a prediction horizon $\Delta T \in \mathbb{R}_+$, which specifies how far ahead in time (from the current time $\tau_t$) the prediction should be made. The prediction module in GRUwE estimates the future output at time $\tau_t + \Delta T$ by conditioning on the Markov state $\mathcal{H}_t$ at time $\tau_t$. It operates in two stages by: (i) projecting the Markov

state to the target time $\tau_t + \Delta T$, and (ii) generating predicted values at the future time using a task-specific output function. For projecting the Markov state to the prediction time one may, in principle, choose arbitrary functions of continuous-time different from the exponential decay functions used to model the effect of elapsed time in the state update (time-triggering reset mechanism). However, for simplicity sake, GRUwE shares the exponential basis functions to model the dynamics of Markov state in the prediction module:

$$\boldsymbol{\gamma}(\boldsymbol{\Delta T}) = \exp\{-\max(\mathbf{0}, \mathbf{W}_\gamma \boldsymbol{\Delta T} + \mathbf{b}_\gamma)\}, \tag{5}$$

$$\mathbf{g}_{t,\Delta T} = \boldsymbol{\gamma}(\boldsymbol{\Delta T}) \odot \mathbf{h}_t. \tag{6}$$

After projecting the state to the prediction time, the prediction module generates the output by applying a task-specific output function $F_{out}(\cdot)$ to the projected state at prediction horizon ($\Delta T$):

$$\widehat{\mathbf{x}}_{t,\Delta T} = F_{out}(\mathbf{g}_{t,\Delta T}). \tag{7}$$

For the next observation prediction, the output function can be defined by using a linear combination of the components of the projected hidden state or a more complex nonlinear combination defined by a Multi-Layer Perceptron (MLP). For simplicity sake, in our experiments, we use a linear combination specified using parameters $\{\mathbf{W}_{out}, \mathbf{b}_{out}\}$ as the $F_{out}(\cdot)$:

$$\widehat{\mathbf{x}}_{t,\Delta T} = \mathbf{W}_{out}\ \mathbf{g}_{t,\Delta T} + \mathbf{b}_{out}. \tag{8}$$

For the next event prediction, we use GRUwE's decayed hidden state ($\mathbf{g}_{t,\Delta T}$) to formulate the Conditional Intensity Function (CIF) for event type $k$ at time $\tau_t + \Delta T$ as:

$$\lambda(\tau_t + \Delta T \mid \mathbf{h}_t, k) = \text{Softplus}(\mathbf{w}_\lambda^k\ \mathbf{g}_{t,\Delta T} + b_\lambda^k). \tag{9}$$

Here, the linear projection parameters $\mathbf{w}_\lambda^k$ and $b_\lambda^k$ are the parameters of the CIF model specific to event type $k$. We apply the Softplus non-linearity to ensure that the output is positive, as intensity represents the instantaneous rate of event arrivals.

**Summary of GRUwE.** Our proposed model GRUwE, is a latent state transition model that maintains a Markov state of the process in continuous time to summarize the historical events. GRUwE primarily consists of a state transition module and a prediction module. To update its Markov state, the state transition module defines: i) time-triggered reset mechanism to account for the time-elapsed, and ii) observation-triggered reset mechanism to incorporate the influence of the observations. GRUwE's prediction module reuses the exponential basis function (used to model inter-arrival times) to support continuous-time prediction module. For more clarity, Figure 1b illustrates the process of sequentially feeding observation and mask vectors along with the time elapses for each time step, followed by a state transition module, to update the previous state as a function of current inputs. Ultimately, prediction module generates a prediction at a future time $\Delta T$ with the help of the updated state.

## 5 Analysis of Exponential Basis Functions

We theoretically analyze the properties and inductive biases introduced by the proposed exponential basis functions for modeling the temporal dynamics.

**Asymptotic Behavior.** We analyze the asymptotic behavior of the exponential basis function $\boldsymbol{\gamma}(\Delta T) = \exp\{-\max(\mathbf{0}, \mathbf{W}_\gamma \Delta T + \mathbf{b}_\gamma)\}$ and its effect on the decayed state $\mathbf{g}_{t,\Delta T} = \boldsymbol{\gamma}(\Delta T) \odot \mathbf{h}_t$. Consider the component-wise behavior for dimension $i$ as $\Delta T \to \infty$:

**Case 1: $W_{\gamma,i} > 0$ (State Reset)**

- **Behavior:** $g_{t,\Delta T,i} \to 0$

- **Explanation:** When $W_{\gamma,i} > 0$, the argument $W_{\gamma,i}\Delta T + b_{\gamma,i} \to \infty$ as $\Delta T$ increases. As a result:

$$\max(0, W_{\gamma,i}\Delta T + b_{\gamma,i}) \to \infty \quad \Rightarrow \quad \exp(-\infty) \to 0.$$

**Case 2:** $W_{\gamma,i} = 0$ **(Constant Decay)**

- **Behavior:** $g_{t,\Delta T,i} \to \exp(-\max(0, b_{\gamma,i})) \cdot h_{t,i}$

- **Explanation:** With zero weight, the decay term simplifies to a constant:

$$\boldsymbol{\gamma}_i(\Delta T) = \exp(-\max(0, b_{\gamma,i})).$$

  This results in a constant decay of the hidden state component. The final value depends on the bias term $b_{\gamma,i}$:

  - If $b_{\gamma,i} \leq 0$: $\exp(-\max(0, b_{\gamma,i})) = \exp(0) = 1$ (no decay).
  - If $b_{\gamma,i} > 0$: $\exp(-\max(0, b_{\gamma,i})) = \exp(-b_{\gamma,i}) < 1$ (constant decay).

**Case 3:** $W_{\gamma,i} < 0$ **(No Decay)**

- **Behavior:** $g_{t,\Delta T,i} \to h_{t,i}$

- **Explanation:** For negative weights, the decay rate $W_{\gamma,i}\Delta T + b_{\gamma,i} \to -\infty$:

$$\max(0, W_{\gamma,i}\Delta T + b_{\gamma,i}) \to 0 \quad \Rightarrow \quad \exp(0) = 1.$$

Asymptotic analysis reveals that the proposed exponential basis functions have three operating modes to express the latent temporal dynamics for each hidden component. Based on the learned value of the parameter $\mathbf{W}_\gamma$, the decay mechanism can either decay to zeros, perform constant decay, or no decay at all. The no decay mode can be used to *remember* the hidden state components indefinitely as time passes. This mechanism can be used to capture very long-term dependencies. The constant-decay mode is useful to capture dependencies that relies on counts, or perhaps when the inter-arrival times are approximately regularly spaced (since the decay is independent of time-elapsed $\Delta T$). When $\mathbf{W}_\gamma > 0$, the exponential basis functions decay with increasing $\Delta T$, causing the state to contract toward the zero vector (or, equivalently to its initial state). Intuitively, it means that as more time passes (since the last observation), the impact of historical arrivals begin to diminish. In the limit, time-triggered reset mechanism forces the state representation to reset to its initial state. This leads to an interesting alternative interpretation of the exponential basis functions: we are effectively learning a *dynamic forgetting process*, in which the *forget rates* are regulated with the help of exponential basis functions.

**Lipschitz Continuity.** The mapping $\gamma : \mathbb{R}_+ \to [0, 1]$ is the exponential decay function defined as $\gamma(\Delta T) = \exp\{-\max(0, W\Delta T + b)\}$, where $W > 0$ and $b \in \mathbb{R}$ are fixed scalar parameters. Then, $\gamma$ is Lipschitz continuous on $\mathbb{R}_+$, with Lipschitz constant:

$$L = W \cdot \exp(-b).$$

Refer to the detailed proof in Appendix 1.1. This result establishes that the exponential basis function $\gamma(\Delta T)$ used in GRUwE is Lipschitz continuous with a closed-form constant that depends directly on the decay weight $\mathbf{W}_\gamma$ and $\mathbf{b}_\gamma$. The decay function is guaranteed not to change faster than $W \cdot \exp(-b)$, enabling GRUwE to dynamically adjust the smoothness of the exponential decay function and ensuring that the function evolves smoothly over time.

## 6 Empirical Evaluation

We evaluate the performance of GRUwE on both next observation and next event prediction tasks.

**Datasets.** For next observation prediction, we use four irregularly sampled multivariate time series datasets: the United States Historical Climatology Network (USHCN), which contains weather-related attributes, and three clinical datasets: PhysioNet, MIMIC-III-Small, and MIMIC-III-Large. Among the four datasets, USHCN is made irregular by randomly masking 50% of the observed time-points as proposed in Schirmer et al. (2022), whereas Physionet, MIMIC-III-Small, and MIMIC-III-Large are irregular in nature. For the event prediction, we use Retweet Zhou et al. (2013) consisting of retweet event sequences; Taxi Whong (2014) consisting of taxi pick-up and drop-off events in five New York city boroughs; Amazon Ni et al. (2019) which

consists of sequences of user provided product reviews; StackOverflow Leskovec & Sosič (2016) consisting of sequence of awards received by users on a Q&A website, and Taobao Xue et al. (2022) consisting of user activities on Taobao platform. We provide the description of the datasets in Appendix 2.

**Models.** For the next observation prediction, we compare the proposed GRUwE models with the SOTA baseline models: GRU-$\Delta_t$ Che et al. (2018), RKN-$\Delta_t$ Becker et al. (2019), GRU-D Chung et al. (2014), mTAND Shukla & Marlin (2021), CRU Schirmer et al. (2022), f-CRU Schirmer et al. (2022), ODE-RNN Rubanova et al. (2019), Latent ODE Rubanova et al. (2019), ContiFormer Chen et al. (2023), T-PatchGNNZhang et al. (2024), GraFITi Yalavarthi et al. (2024). In the next event prediction task, we compare GRUwE with Recurrent CIF models: RMTPP Du et al. (2016), NHPMei & Eisner (2017); Attention-based CIF approaches including THP Zuo et al. (2020), SAHP Zhang et al. (2020), ATTNHP Yang et al. (2022) and Cumulative CIF approach FullyNN Omi et al. (2019). We include more details on the implementation of the baselines in the Appendix 3).

**Model training.** For the next observation prediction task, the models processes a masked multivariate time series input, where the future observation are omitted. Models are optimized to output the entire input sequence after observing partially masked sequence. We use Mean Squared Error (MSE) loss function to optimize the model parameters. Concretely, for a sequence $s$ with $N$ masked time steps, prediction horizon $\Delta T$, observation mask vector $\mathbf{m}_{t+\Delta T}$, prediction values $\hat{\mathbf{x}}_{t,\Delta T}$ and observed values $\mathbf{x}_{t,\Delta T}$ , we optimize the model parameters by minimizing the masked MSE loss:

$$\mathcal{L}_{\text{obs}}(s) = \frac{1}{\sum_{i=1}^{N} \mathbf{m}_{t+\Delta T}^{(i)}} \sum_{i=1}^{N} \mathbf{m}_{t+\Delta T}^{(i)} \cdot \left( \hat{\mathbf{x}}_{t,\Delta T}^{(i)} - \mathbf{x}_{t,\Delta T}^{(i)} \right)^2 . \tag{10}$$

For the event prediction task, conditional intensity function parameterized with the help of GRUwE is optimized by maximizing the log-likelihood of observing a sequence $s = \{(\tau_j, k_j)\}_{j=1}^{L}$ at observation times $\{\tau_1, \tau_2, ..., \tau_L\} \in [0, T]$:

$$\mathcal{L}_{event}(s) = \sum_{j=1}^{L} \log \lambda(t_j \mid \mathcal{H}_j, k_j) - \int_{t=0}^{T} \lambda(t \mid \mathcal{H}_t) \, dt. \tag{11}$$

where the total intensity function is given by: $\lambda(t \mid \mathcal{H}_t) = \sum_{k=1}^{K} \lambda(t \mid \mathcal{H}_t, k)$.

**Evaluation criteria.** For the next observation prediction, similar to prior works Schirmer et al. (2022); Yalavarthi et al. (2024); Zhang et al. (2024) we evaluate models on MSE and Mean Absolute Error (MAE) on the masked future values on the sequences in the test split. Using standard evaluation metrics in event prediction literature Yang et al. (2022); Zuo et al. (2020); Zhang et al. (2020), we compare the model quality on the test set using: RMSE: Root Mean Squared Error in predicting the next event arrival time; ER: short for Error Rate, is the percentage of incorrectly predicted next event type and the log-likelihood of the next observed event. To ensure robust evaluation, we conduct experiments using 5 distinct random seeds and report the mean and standard deviation of the performance metrics.

**Hyperparameter search.** We conduct model-specific hyperparameter tuning by training various model configurations on the training set and evaluating their performance on the validation set. Hyper-parameter ranges for the methods were adjusted to include the optimal hyper-parameter ranges reported by the original authors. For each model, we select the configuration that achieves the lowest validation MSE (for forecasting) and lowest validation LL (for event prediction) and report its test performance. We include more details on the hyperparameter tuning in Appendices 4 to 9.

## 7 Results and Discussion

We evaluate the proposed GRUwE model on next observation prediction and next event prediction tasks.

**Next observation prediction.** We evaluate the performance of GRUwE and baselines by predicting the next observation for each variable defining the multivariate time series given the history of past observations. For the Physionet dataset, the model observes the first 24 hours of the total 48-hour period to predict the next observations for all (37) variables. Similarly, for the USHCN dataset, with daily samples over four years, we use the first half to predict the next observation for all 5 variables. For the MIMIC-III-Large

| Model | USHCN | | Physionet | | MIMIC-III-Large | | MIMIC-III-Small | |
|---|---|---|---|---|---|---|---|---|
| | MSE $\times 10^{-2}$ | MAE $\times 10^{-2}$ | MSE $\times 10^{-2}$ | MAE $\times 10^{-2}$ | MSE $\times 10^{-2}$ | MAE $\times 10^{-2}$ | MSE $\times 10^{-1}$ | MAE $\times 10^{-1}$ |
| f-CRU | 0.020± 0.007 | 0.455± 0.077 | 1.095± 0.069 | 5.295± 0.132 | 1.191± 0.043 | 6.535± 0.214 | 7.808± 0.021 | 5.963± 0.029 |
| mTAND | 0.007± 0.004 | 0.194± 0.069 | 0.330± 0.015 | 3.411± 0.130 | 1.260± 0.018 | 6.885± 0.082 | 5.960± 0.003 | 5.087± 0.017 |
| GRU-D | 0.015± 0.009 | 0.386± 0.110 | 0.672± 0.026 | 5.459± 0.094 | 0.984± 0.028 | 5.853± 0.102 | 6.621± 0.012 | 5.486± 0.032 |
| Latent ODE | 0.007± 0.004 | 0.127± 0.031 | 0.676± 0.005 | 5.302± 0.001 | 1.209± 0.018 | 6.490± 0.049 | 6.658± 0.010 | 5.547± 0.026 |
| ContiFormer | 0.005 ± 0.002 | 0.125 ± 0.016 | 0.479± 0.024 | 4.232± 0.001 | 1.348± 0.093 | 6.941± 0.399 | 7.119± 0.022 | 5.873± 0.035 |
| ODE-RNN | 0.019± 0.017 | 0.220± 0.116 | 0.770± 0.042 | 5.519± 0.273 | 1.429± 0.046 | 7.251± 0.262 | 6.401± 0.008 | 5.302± 0.017 |
| CRU | 0.030± 0.019 | 0.519± 0.166 | 0.807± 0.035 | 5.233± 0.242 | 1.236± 0.035 | 6.735± 0.139 | 6.927± 0.018 | 5.812± 0.030 |
| RKN-$\Delta_t$ | 0.015± 0.016 | 0.367± 0.178 | 0.680± 0.042 | 4.854± 0.197 | 1.292± 0.042 | 6.820± 0.081 | 8.205± 0.025 | 6.185± 0.043 |
| GRU-$\Delta_t$ | 0.035± 0.001 | 0.717± 0.018 | 0.449± 0.018 | 4.160± 0.154 | 1.414± 0.051 | 7.226± 0.086 | 6.863± 0.019 | 5.617± 0.031 |
| T-PatchGNN | 0.065± 0.031 | 0.909± 0.405 | 0.338± 0.032 | 3.259± 0.168 | 1.226± 0.010 | 6.689± 0.151 | 5.664± 0.004 | 4.777± 0.016 |
| GraFITi | 0.074± 0.015 | 0.673± 0.042 | 0.233± 0.009 | 2.781 ± 0.025 | 1.419± 0.032 | 7.448± 0.121 | 4.692± 0.003 | 4.287 ± 0.021 |
| HyperIMTS | 0.052± 0.013 | 0.539± 0.021 | 0.269± 0.003 | 2.932± 0.020 | OOM | OOM | 4.734± 0.008 | 4.364± 0.015 |
| GRUwE | 0.015± 0.013 | 0.377± 0.187 | 0.261± 0.006 | 2.957± 0.035 | 0.816 ± 0.026 | 5.278 ± 0.036 | 4.651 ± 0.015 | 4.296± 0.022 |

Table 1: **Next observation prediction.** Comparison of models on next observation prediction on USHCN, Physionet MIMIC-III-Small and MIMIC-III-Large datasets. We report the mean and standard deviation of MSE and MAE on five distinct random seeds. "OOM" refers to out-of-memory GPU error triggered during training.

dataset, models consider values of 506 variables over past 48 hours from a randomly sampled time point in the patient record. The models are compared on predicted values for the next observations made on 363 numerical variables representing vital signs and labs. For the MIMIC-III-Small dataset, the model observes the first half of the 96 numerical variables to predict their next observation. Table 1 presents the predictive performance of all models on the next observation prediction task across the four datasets, evaluated over multiple random seeds. Next we briefly summarize the results on individual datasets below. On the USHCN dataset, ContiFormer, Latent ODE, mTAND are the top performing models in terms of MSE and MAE. On the Physionet dataset, GraFITi, GRUwE and HyperIMTS are the top performing methods. On the MIMIC-III-Large dataset, GRUwE achieves the lowest MSE and MAE, outperforming all competing models, with GRU-D ranking second among the baselines. GRU-D uses a mean-reverting interpolation scheme for missing inputs, an assumption that likely holds for this setting, which we believe is contributing to its competitive predictive accuracy. We note that MIMIC-III-Large is the most challenging dataset of all the datasets used: it consists of high-dimensional input observations and highly varied observation sequence lengths as reported in Table 4. On the MIMIC-III-Small dataset, GRUwE outperforms all the baseline models. GraFITi and HyperIMTS remain competitive.

**Missing data perspective.** Prior work Singh (1997); Ramoni & Sebastiani (1997) categorizes missingness into three types: (i) Missing Completely at Random (MCAR), (ii) Missing at Random (MAR), and (iii) Not Missing at Random (NMAR). In our experiments, the USHCN dataset corresponds to the MCAR setting, where missingness is synthetically introduced by randomly removing observations. In contrast, the Physionet and MIMIC-III datasets are instances of NMAR, where missingness is contextual, i.e., *dependent* on observed and unobserved values. For example, certain laboratory tests may only be ordered after abnormal vital signs are detected. Our results indicate that while GRUwE remains competitive under MCAR, it performs particularly well under NMAR conditions, outperforming all compared models on two out of three datasets. This suggests that GRUwE more effectively captures dependencies inherent in the data-generating process.

**Next event prediction.** We report the comparison of the TPP models in the Table 2 and plot the log-likelihoods in Figure 2. Notably, the FullyNN model Omi et al. (2019) is designed for single event prediction. While RMSE and LL metrics can be computed, the ER metric cannot be evaluated for this model in our comparison. As demonstrated in Figure 2, GRUwE outperforms all intensity-based models in terms of log-likelihood comparison, except on the Amazon dataset, where it achieves the second-highest performance. For the Taxi dataset, which has the least number of training sequences across all datasets, GRUwE ranks first in terms of the RMSE and ER. If we compare among all RNN-based intensity models (i.e. GRUwE, NHP, RMTPP), GRUwE demonstrates the best performance on all metrics and datasets showing the effectiveness of the proposed approach.

**Rank analysis**. Overall, our results in the Table 2 indicate that GRUwE is consistently among the top performing models across all the datasets and evaluation metrics. To assess its benefits more rigorously, we

---

[1]Original FullyNN model does not support multi-type event.

| Model | Taxi RMSE(↓) | Taxi ER(↓) | Retweet RMSE(↓) | Retweet ER(↓) | StackOverflow RMSE(↓) | StackOverflow ER(↓) | Amazon RMSE(↓) | Amazon ER(↓) | Avg. Rank (↓) |
|---|---|---|---|---|---|---|---|---|---|
| FullyNN [1] | 0.373± 0.005 | N.A. | **21.92** ±0.159 | N.A. | 1.375±0.015 | N.A. | 0.615±0.005 | N.A. | 4.75 |
| NHP | **0.369** ± 0.001 | 9.22±0.005 | 22.32± 0.001 | **40.25** ± 0.002 | 1.369±0.001 | 55.78±0.007 | 0.612±0.001 | 68.30±0.009 | 2.88 |
| A-NHP | 0.370± 0.000 | 11.42±0.016 | 22.28±0.018 | 41.05±0.004 | 1.370±0.000 | 55.51±0.001 | 0.612±0.000 | **65.65** ±0.002 | 3.38 |
| THP | **0.369** ± 0.000 | 8.85± 0.003 | 22.32±0.001 | **40.25** ±0.002 | **1.368** ±0.002 | 55.60±0.003 | 0.612±0.000 | 66.72±0.009 | 2.13 |
| SAHP | 0.372±0.003 | 9.75± 0.001 | 22.40±0.301 | 41.60±0.002 | 1.375±0.013 | 56.10±0.005 | 0.619±0.005 | 67.70±0.006 | 5.38 |
| RMTPP | 0.370±0.000 | 9.86±0.009 | 22.31±0.209 | 44.10±0.003 | 1.370±0.001 | 57.50±0.000 | 0.634±0.011 | 73.66±0.052 | 5.25 |
| GRUwE | **0.369** ±0.000 | 8.58 ±0.003 | 22.21±0.134 | 40.81±0.004 | 1.369±0.001 | 55.34 ±0.004 | **0.611** ±0.001 | 67.24±0.004 | **1.75** |

Table 2: **Next event prediction.** Model performance comparison for the next event prediction on the Taxi, Retweet, StackOverflow and Amazon datasets. We report the mean and standard deviation of the metrics on five distinct random seeds. For RMSE and ER metrics, lower is better.

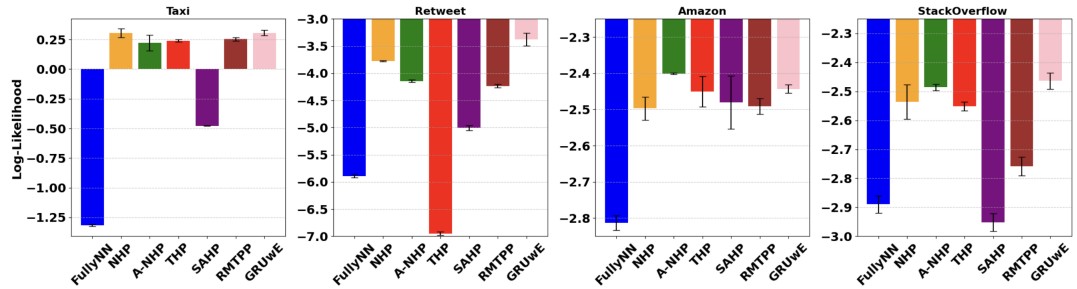

Figure 2: Comparison of Log-Likelihood (LL) for all the TPP models on next event prediction task. A higher value of LL indicates a better fit.

summarize the results by adding a new rank-based metric that calculates the average rank of each model across all datasets for both RMSE and ER metrics. The ranking results show that GRUwE achieves the highest rank score of **1.75**, while the second-best THP model has the score 2.13. More details on this rank analysis is provided in the Appendix 10.

**Differences to GRU-D Che et al. (2018).** GRUwE is most closely related to the previously proposed GRU-D model since it also combines GRU and exponential decay functions. However, the following key differences remain. While GRU-D imputes the input missing values by learning exponential schedules to revert to its empirical mean, GRUwE lifts this mean-reverting assumption by applying zero imputation to handle missingness. This reduces cascading errors in the final prediction caused by imputation errors. GRU-D was proposed for classification, and it is unclear how it can be adapted to perform continuous-time predictions. GRUwE addresses this by introducing a flexible prediction module. GRUwE simplifies GRU-D's Markov state representation significantly by removing the need to store and update: i) the last observed variable values, ii) their respective time elapsed since observation, and iii) empirical mean for each variable. It is important to highlight that all of these differences make GRUwE a simpler model compared to GRU-D, while significantly boosting the predictive accuracy across all datasets in the next observation prediction benchmark (Table 1).

**Computational cost analysis.** Beyond evaluating predictive performance, we assess the computational requirements of each model. Figure 3 summarizes key metrics, including training time, peak memory usage during training, and inference time on retrospective test data. For a focused analysis of deployment efficiency, Figure 4a compares the average inference time and memory consumption of top-performing models in an online (sequential) setting. A comprehensive discussion on computational complexity is provided in Appendix 11.

Regarding training time on retrospective data (Figure 3a), methods parallelizable over the time dimension (T-PatchGNN, mTAND, GraFITi, HyperIMTS) are fastest. These are followed by RNN-based models (GRUwE, GRU-D, GRU-$\Delta_t$), then models with linear dynamics (RKN-$\Delta_t$, CRU, f-CRU). Lastly, models that require invoking numerical solvers (ContiFormer, Latent ODE, ODE-RNN) consume the most amount of train time. As Figure 3b indicates, this speed often entails a trade-off: time-parallelizable models achieve shorter training times at the cost of substantially higher memory consumption. For inference on retrospec-

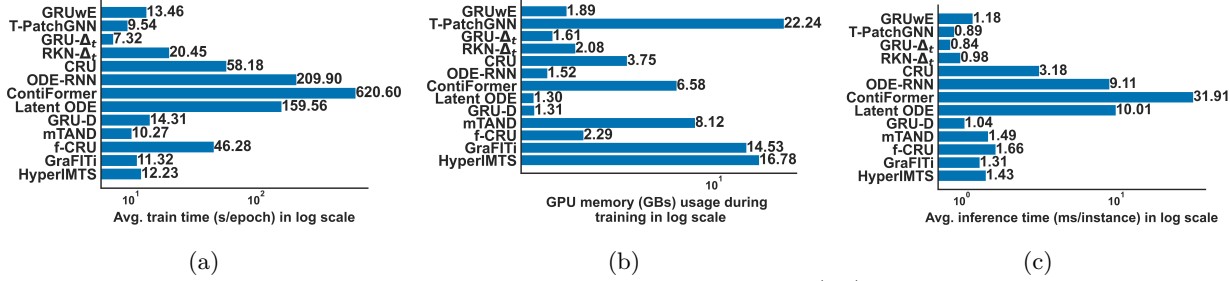

(a)                                        (b)                                        (c)

Figure 3: **Computational cost analysis on retrospective data.** (**3a**) compares the train times for all models; (**3b**) compares the peak GPU memory usage during training; (**3c**) compares the inference times on retrospective data. Lower is better for all plots.

tive data where models process the full history at once, most approaches exhibit comparable inference times, with ODE-based methods being notable exceptions (Figure 3c). The computational profile changes significantly in an online setting, where data arrive sequentially. Here, models that require the full observation history must buffer and reprocess all past data with each new prediction. This leads to growing memory and computational demands over time, as sequence length increases, creating a computational bottleneck. In contrast, GRUwE maintains a compact Markov state updated incrementally, enabling highly efficient sequential inference. As a result, as shown in Figure 4a, GRUwE incurs significantly lower computational overhead during online deployment.

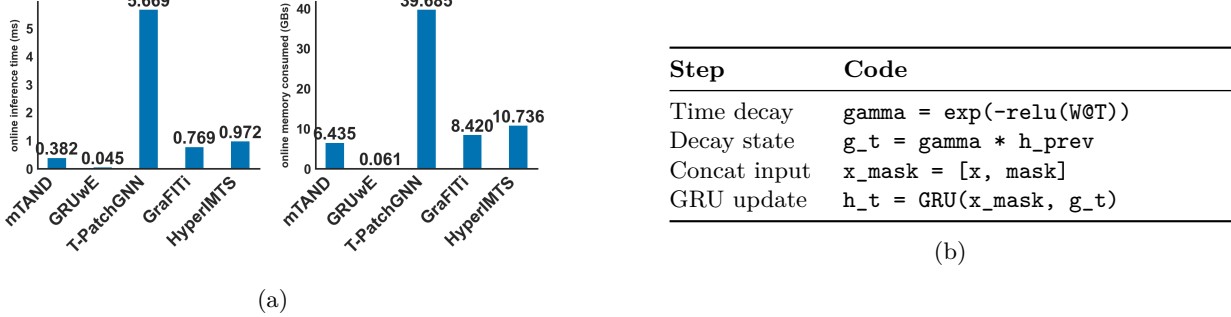

| Step | Code |
|---|---|
| Time decay | `gamma = exp(-relu(W@T))` |
| Decay state | `g_t = gamma * h_prev` |
| Concat input | `x_mask = [x, mask]` |
| GRU update | `h_t = GRU(x_mask, g_t)` |

(a)                                                                                    (b)

Figure 4: (**a**) Comparison of inference time (**left**) and memory consumption (**right**) in online deployment. Lower is better in both cases. (**b**) GRUwE's step function pseudocode.

**Easy to implement.** As shown in Figure 4b, GRUwE's reset state mechanisms can be implemented in just a few lines of code as it builds on the components that are natively supported by most modeling packages. An easy model implementation is preferred not only because it improves readability, but also reduces the scope for introducing bugs, making the model easier to port, reproduce, debug, and maintain.

**GRUwE ablation analysis.** To better assess the contribution of the proposed GRUwE model, we conduct a series of architectural ablation studies. Specifically, we investigate: (i) how GRUwE compares to variants that employ alternative time basis functions, such as learnable linear (GRU-Lin) or sinusoidal (GRU-Sin) basis; (ii) how the predictive performance changes when model complexity is increased by using *separate* learnable exponential function for the transition and prediction functions (GRUwE-DualExp); and (iii) how performance is affected when model complexity is reduced by *freezing* the randomly initialized exponential basis (GRUwE-FrozenExp) during model training. Across all datasets (as presented in Table 3), GRUwE consistently matches or outperforms its ablated variants, highlighting the importance of a shared, learnable exponential basis. Replacing the exponential basis with linear or sinusoidal functions degrades performance significantly. This supports our hypothesis that exponentials provide a better inductive bias, as their memoryless, multiplicative form makes temporal effects naturally composable over arbitrary time gaps. Learning separate exponential basis for the transition and prediction functions helps in one but does not outperform GRUwE in most cases, suggesting that sharing a single temporal basis regularizes the training and yields a more robust temporal representation. Lastly, reducing the complexity by freezing the exponential basis

| Model | USHCN (MSE $\times 10^{-2}$) | Physionet (MSE $\times 10^{-2}$) | MIMIC-III-Large (MSE $\times 10^{-2}$) | MIMIC-III-Small (MSE $\times 10^{-1}$) |
|---|---|---|---|---|
| GRU-Lin | 0.019 | 0.925 | 1.532 | 7.573 |
| GRU-Sin | 0.016 | 0.359 | 1.216 | 7.324 |
| GRUwE-DualExp | **0.013** | 0.269 | 0.854 | 4.670 |
| GRUwE-FrozenExp | 0.016 | 0.256 | 0.848 | 4.665 |
| GRUwE | 0.014 | **0.253** | **0.831** | **4.650** |

Table 3: **GRUwE ablation comparison**. GRUwE variants modify the time basis function: using linear or sinusoidal functions, learning separate basis for transition and prediction, and freezing the randomly initialized exponential basis.

yields competitive but worse performance in all cases, indicating that adapting decay rates to the task is important.

**Limitations.** Unlike compared baselines such as mTAND and CRU, our proposed architecture, GRUwE, doesn't have a built-in notion of uncertainty, and we consider incorporating it in the future work will be an exciting extension of our current work.

## 8 Conclusion

In summary, this work revisits the challenge of modeling irregularly sampled multivariate time series through the lens of simplicity and computational efficiency. By introducing GRUwE, we demonstrate that a principled extension of the GRU architecture, augmented with learnable exponential basis functions, can achieve state-of-the-art predictive performance across both continuous-time forecasting and event prediction tasks, while remaining significantly computationally efficient during inference in the online deployment. These results demonstrate that carefully designed recurrent architectures grounded in classical principles, can compete or outperform more complex neural and differential equation–based models. We believe this work provides useful guidance for future research on practical and scalable modeling of irregular time series.

## 9 Acknowledgement

The research work in this paper was supported in part by NIH grant R01 EB032752, and by the University of Pittsburgh Center for Research Computing and Data resource RRID:SCR 022735 using HTC cluster, which is supported by NIH award S10OD028483. The content of this paper is solely the responsibility of the authors and does not necessarily represent the official views of the NIH.

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

# Appendix

# Table of Contents

## 1 Theory

### 1.1 Lipschitz Continuity of Exponential Decay Function

**Theorem 1.1 (Lipschitz Continuity of Exponential Decay Function).** *Let* $\gamma : \mathbb{R}_+ \to \mathbb{R}$ *be the exponential decay function defined as*

$$\gamma(\Delta T) = \exp\left\{-\max\left(0, W\Delta T + b\right)\right\},$$

*where* $W > 0$ *and* $b \in \mathbb{R}$ *are fixed scalar parameters. Then,* $\gamma$ *is Lipschitz continuous on* $\mathbb{R}_+$, *with Lipschitz constant*

$$L = W \cdot \exp(-b).$$

*Proof.* We first compute the derivative of $\gamma$ with respect to $\Delta T$ in piecewise form:

$$\frac{d}{d\Delta T}\gamma(\Delta T) = \begin{cases} 0, & \text{if } \Delta T \leq -\frac{b}{W}, \\ -W \cdot \exp\left(-W\Delta T - b\right), & \text{if } \Delta T > -\frac{b}{W}. \end{cases}$$

Observe that $\gamma$ is continuous everywhere and differentiable almost everywhere (it is non-differentiable only at the point $\Delta T = -\frac{b}{W}$). The function $\gamma$ is Lipschitz continuous if its derivative is bounded almost everywhere, and the essential supremum of the absolute value of the derivative gives the Lipschitz constant.

For $W\Delta T + b > 0$, the magnitude of the derivative is:

$$\left|\frac{d\gamma}{d\Delta T}\right| = W \cdot \exp(-W\Delta T - b).$$

The maximum of this expression over $\Delta T \in \mathbb{R}_+$ occurs at $\Delta T = 0$, giving:

$$\left|\frac{d\gamma}{d\Delta T}\right| \leq W \cdot \exp(-b).$$

For $\Delta T < -\frac{b}{W}$, the derivative is zero. Therefore, the global Lipschitz constant is:

$$L = \sup_{\Delta T \geq 0}\left|\frac{d\gamma}{d\Delta T}\right| = W \cdot \exp(-b).$$

$\square$

## 1.2 Monotonic Latent Dynamics

**Remark 1** (**Exponential Basis Functions Induces Growth and Decay**). *Exponential basis functions can induce exponentially growing and decaying trends in the state as a function of time elapsed.*

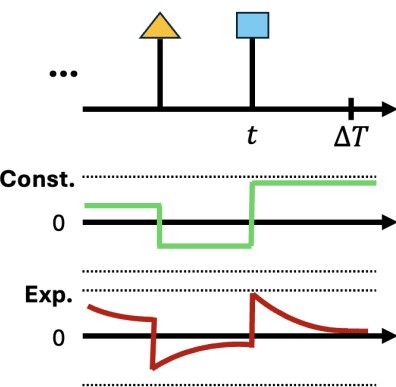

Figure 5: An abstract diagram of latent dynamics in the case of GRU (**top**) and GRUwE (**bottom**) highlighting that GRUwE can recover monotonic latent dynamics between two consecutive event or observation arrivals.

We note that $\boldsymbol{\gamma(\Delta T)} \in (0,1]^D$, the $i^{th}$ component of state is bounded, i.e., $h_{t,i} \in [-1,1]$. Since $\mathbf{g}_{t,\Delta T} = \boldsymbol{\gamma(\Delta T)} \odot \mathbf{h}_t$, if $h_{t,i} \in [-1,0)$, then $h_{t+\Delta T,i} \geq h_{t,i}$. Basically, the transformed state value is an upper-bound of the initial value if it is negative. If the initial value is positive, then the transformed value is a lower-bound: i.e., $h_{t,i} \in (0,1]$, then $h_{t+\Delta T,i} \leq h_{t,i}$. Therefore, depending on the state $\mathbf{h}_t$, the exponential basis function update can cause the state value to grow or decay exponentially. This property makes exponential basis functions highly expressive for time-series modeling.

## 2 Datasets

### 2.1 USHCN

United States Historical Climatology Network (USHCN) Menne et al. (2016) is a publicly available dataset (`https://data.ess-dive.lbl.gov/view/doi%3A10.3334%2FCDIAC%2FCLI.NDP019`) consisting of daily measurements of 5 meteorological variables including min temperature, max temperature, precipitation, snowfall, and snow depth from 1218 observing stations across the United States. It is worth noting that there are multiple versions of datasets extracted from USHCN that have been used to evaluate irregularly sampled tim series. We extract the dataset in our analysis using the steps mentioned in Schirmer et al. (2022). We mimic their data pre-processing pipeline by: i) sub-selecting 1168 stations over a 4-year period ranging from 1990 to 1993 ii) subsample 50% of time-points to increase irregularities in time dimension; and setting unobserved rate to 20% to increase sparsity of observations, iii) 20% of the entire dataset is used for testing; we train and validate on the remaining 80% data; 25% of that is used for validation.

To be able to compare to methods listed in the Schirmer et al. (2022) for the prediction task, we replicate their: (1) pre-processing logic; (2) splitting of dataset into train, validation and test sets by using the same seed; (3) 20% partial observability in feature dimension and 50% in time dimension in GRUwE run; (4) masking logic for prediction task.

### 2.2 Physionet

Predicting Mortality of ICU Patients: The PhysioNet/Computing in Cardiology Challenge 2012 Silva et al. (2012) made publicly available (`https://physionet.org/files/challenge-2012/1.0.0/`) 8000 ICU patient stays that span 48 hours reporting 37 clinical real-valued time series variables observed at irregular time-intervals. The dataset includes various variables, including Non-Invasive Mean Arterial Pressure, Platelets, Sodium, and several others. We follow the pre-processing of this dataset described by Schirmer et al. (2022). Observations in time are rounded by 6 minutes. Similar to USHCN, test split consists of 20% of the data, the rest is used for training and validation; validation set consists of 25% of this split.

For fair comparison to methods listed in Schirmer et al. (2022), we ensure (1) 6-minute quantization (as done in Rubanova et al. (2019); Schirmer et al. (2022)); (2) splits are created using the same seeds; (3) same masking logic is applied for prediction task.

### 2.3 MIMIC-III-Large

In the following, we outline the steps to extract MIMIC-III-Large dataset from the MIMIC-III Johnson et al. (2023) (`https://physionet.org/content/mimiciii/1.4/`). A population cohort of 10,265 hospital admissions (on 8,799 patients) are extracted from MIMIC-III based on the following criteria: i) patient record is recorded in MetaVision critical care information system, ii) the length of the patient record is between 2 and 20 days, iii) the age of the patient is between 18 and 90. From all EHR tables available in MIMIC-III, we extract (irregularly sampled) time series that include vital signs (such as Heart Rate and Mean Arterial Pressure), lab results (such as Glucose and Hemoglobin), administered medications (such as Propofol and Norepinephrine), and procedures (such as Intubation). The vital signs and lab results are numerical time series, while the rest are indicator time series, indicating if and when the event occurred.

We filter out any univariate time series that occurs less than 500 times across all patients in the cohort resulting in total of 506 time series: 393 numerical (vitals and labs), 77 medications and 36 procedures event time series. We define an *A-point*, abbreviated for Anchor-point, as a temporal moment at which a decision-making system can formulate a prediction based on the past sequence of events. We extract A-points from the filtered patient records regularly with frequency of 24 hours i.e. one sample is extracted every 24 hours from the patient record. We standardize all real-valued univariate time series (i.e. vital signs and labs) data using min-max scaling; and encode all other indicator time series as binary value 0/1. Value is 1 if event occurs; 0 otherwise. 80% of the patient hospital admission are used for training and validation (20% of train split); the rest is used for testing. Splits are constructed on disjoint patients.

Additional pre-processing is required to remove missing values encoded as 9999999 in the numerical time series. To remove outliers from univariate time series (for example, very large values of the order of $1e5$),

we filter out observations that fall either in $< 0.1$ or $> 99.9$ percentile ranges. Subsampling of A-points is performed as follows:

1. For each split:

   a. For each patient admission record, filter out A-points with less than 50 events in history and prediction window.

   b. Next, randomly sample one A-point from the filtered A-points i.e. one sample per patient admission record.

2. Subsample 1000 A-points from train, 250 from validation and 200 from test set to be used for experimentation.

## 2.4 MIMIC-III-Small

We adopt the data extraction methodology described in De Brouwer et al. (2019) to derive the MIMIC-III-Small dataset from MIMIC-III Johnson et al. (2023). Each sample comprises 96 time series variables, irregularly observed over a 48-hour long patient visit. For a comprehensive overview of the dataset preprocessing and extraction steps, we refer readers to Appendix K in De Brouwer et al. (2019).

## 2.5 Dataset Attributes

Table 4 summarizes the mean, standard deviation, minimum and maximum of all the sequences per dataset. Importantly, the MIMIC-III dataset does not undergo any time discretization, resulting in sequence lengths spanning from 105 to 4299. In terms of average length, the order is as follows: USHCN > MIMIC-III-Large > Physionet > MIMIC-III-Small.

Table 4: Sequence length statistics and number of target variables across datasets

| Dataset | Sequence length statistics | | | # Target Variables |
|---|---|---|---|---|
| | Mean $\pm$ S.D. | Min | Max | |
| USHCN | $730.0 \pm 0.00$ | 730 | 730 | 5 |
| Physionet | $72.16 \pm 20.93$ | 1 | 185 | 37 |
| MIMIC-III-Small | $48.15 \pm 69.92$ | 2 | 614 | 96 |
| MIMIC-III-Large | $244.30 \pm 288.85$ | 105 | 4299 | 393 |

## 2.6 Event Prediction Datasets

We provide descriptions of the four real-world publicly available datasets used in the next even prediction benchmark of the paper. We summarize the properties of these datasets after providing the descriptions:

- **Amazon:** This dataset includes time-stamped user product reviews behavior from January, 2008 to October, 2018. Each user has a sequence of produce review events with each event containing the timestamp and category of the reviewed product, with each category corresponding to an event type. We work on a subset of 5200 most active users with an average sequence length of 70 and then end up with K = 16 event types

- **Retweet:** This dataset contains time-stamped user retweet event sequences. The events are categorized into K = 3 types: retweets by "small," "medium" and "large" users. Small users have fewer than 120 followers, medium users have fewer than 1363, and the rest are large users. We work on a subset of 5200 most active users with an average sequence length of 70.

- **Taxi:** This dataset tracks the time-stamped taxi pick-up and drop-off events across the five boroughs of the New York City; each (borough, pick-up or drop-off) combination defines an event type, so there are K = 10 event types in total. We work on a randomly sampled subset of 2000 drivers and each driver has a sequence. We randomly sampled disjoint train, dev and test sets with 1400, 200 and 400 sequences.

- **StackOverflow:** This dataset has two years of user awards on a question-answering website: each user received a sequence of badges and there are K = 22 different kinds of badges in total. We randomly sampled disjoint train, dev and test sets with 1400, 400 and 400 sequences from the dataset.

Table 5 provides a description of number of unique event types ($K$), the number of events and sequence length statistics across all four datasets used in our experimental evaluation.

Table 5: Statistics of the used datasets for evaluation

| Dataset | $K$ | # of Event Tokens | | | Sequence Length | | |
|---|---|---|---|---|---|---|---|
| | | Train | Dev | Test | Min | Mean | Max |
| Retweet | 3 | 369000 | 62000 | 61000 | 10 | 41 | 97 |
| Amazon | 16 | 288000 | 12000 | 30000 | 14 | 44 | 94 |
| Taxi | 10 | 51000 | 7000 | 14000 | 36 | 37 | 38 |
| StackOverflow | 22 | 90000 | 25000 | 26000 | 41 | 65 | 101 |

## 3 Baselines

### 3.1 Baselines for Next Observation Prediction Task

We include the following baseline methods to compare against our proposed model GRUwE on the next observation prediction task.

**GRU-$\Delta_t$:** We consider recurrent models GRU-$\Delta_t$ as our baselines in the comparison. Gated Recurrent Unit Chung et al. (2014) have been proposed to model sequences using a set of parametric update equations. Since GRU is not time-aware, a variant of GRU that also feeds in time elapsed (since the last time-step) along with the input GRU-$\Delta_t$ is used in the comparison.

**RKN-$\Delta_t$:** Recurrent Kalman Networks (RKN) Becker et al. (2019) have been proposed to incorporate uncertainty in time series modeling. Similar to GRU-$\Delta_t$, we include the baseline RKN-$\Delta_t$ that includes time elapsed as an additional input to the model.

**GRU-D:** We compare our method to Gated Recurrent Unit-Decay (GRU-D) Che et al. (2018) that uses a mean-reverting imputation function for missing variables; applies learnable exponential decays in the input and latent dimension to account for irregular observed times.

**mTAND:** We evaluate our performance against the Encoder-Decoder generative model, Multi-Time Attention Network (mTAND-Full) Shukla & Marlin (2021), which defines reference points and represents continuous time-points with learnable embeddings to encode their relationship and generate predictions.

**CRU and f-CRU:** We consider Continuous Recurrent Units (CRU) and fast-Continuous Recurrent Units (f-CRU) Schirmer et al. (2022) as our baselines for prediction tasks. f-CRU is a fast implementation of jointly proposed CRU method. CRU consists of an encoder-decoder framework where the hidden state progression is governed by linear stochastic differential equation that allow incorporating arbitrary time-intervals between observations.

**ODE-RNN and Latent ODE:** In our analysis, we consider Latent ODE and ODE-RNN proposed in Rubanova et al. (2019) as comparative baselines. ODE-RNN model consist of latent states that adhere to ODE between observations and are updated at observations using standard RNN update equations. Latent ODE model adopts variational auto-encoder framework wherein the hidden state posterior is modeled by an ODE-RNN model. We used the configuration of Latent ODE with ODE-RNN encoder.

**ContiFormer**: We incorporate ContiFormer Chen et al. (2023) as one of our baselines for the prediction tasks. It builds upon the original transformer architecture by first extending the input irregular data to the continuous-time latent representation by assuming that the underlying dynamics are governed by the ODEs.

**T-PatchGNN**: We include T-PatchGNN Zhang et al. (2024) as one of our baselines for the prediction tasks. T-PatchGNN first segments each time series into patches of uniform temporal resolution followed by the transformer and time-adaptive GNNs to capture dependencies in the multivariate time series. We use the official T-PatchGNN code made publicly available here: `https://github.com/usail-hkust/t-PatchGNN` in our experiment pipeline.

**GraFITi**: We add GraFITi Yalavarthi et al. (2024) as one of our baselines for comparisons on the next observation prediction task. GraFITi casts the time series prediction task in terms of edge weight prediction problem after converting the time series to a sparse graph structure. To incorporate this method, we make use of the official implementation available here: `https://github.com/yalavarthivk/GraFITi`.

**HyperIMTS:** We include HyperIMTS Li et al. (2025) as one of baselines for the next observation prediction task. HyperIMTS models the irregular time series by constructing a hypergraph from observations and learning dependencies with the help of hyperedges connecting the observation nodes. We use the official implementation made available by the authors from this site: `https://github.com/Ladbaby/PyOmniTS`.

## 3.2 Baselines for Event Prediction Task

We include the following temporal point process baselines for event prediction benchmark.

**FullyNN**: We add Fully Neural Network (FullyNN) Omi et al. (2019) to our comparison on the next event prediction task. FullyNN learns a cumulative hazard function for modeling the point process intensities. We use the publicly available implementation of this method from here: `https://github.com/ant-research/EasyTemporalPointProcess`

**NHP**: We include Neural Hawkes Process (NHP) Mei & Eisner (2017) as one of our baselines for the next event prediction task. NHP develops a continuous-time intensity model using the LSTM architecture. We use the publicly available implementation of this method from here: `https://github.com/ant-research/EasyTemporalPointProcess`

**A-NHP**: We include Attention-Neural Hawkes Process (A-NHP) Yang et al. (2022) as a baseline model for next event prediction task. A-NHP with the help of attention mechanism further refines the NHP architecture to define novel intensity functions. Official code for this method is publicly available at `https://github.com/yangalan123/anhp-andtt`. We used the adapted versions from: `https://github.com/ant-research/EasyTemporalPointProcess`.

**THP**: Transformer Hakwes Process (THP) Zuo et al. (2020) is included in the comparison for next event prediction task. THP defines the intensity function using the transformer architecture. We use the publicly available implementation of this model from: `https://github.com/ant-research/EasyTemporalPointProcess`.

**SAHP**: We add Self-Attention Hawkes Process (SAHP) Zhang et al. (2020) to the next event prediction comparison. SAHP uses self-attention mechanism in tranformers to define intensity function. We use the publicly available implementation of this model from: `https://github.com/ant-research/EasyTemporalPointProcess`.

**RMTPP**: Recurrent Marked Temporal Point Process (RMTPP) Du et al. (2016) defines a conditional intensity model based on RNN architecture. We include it in the next event prediction benchmark. We use the code available for this method in the public repository: `https://github.com/ant-research/EasyTemporalPointProcess`.

## 4 Generic hyperparameters for Next Observation Prediction Task

The following hyperparameters are applicable broadly across our forecasting experiments:

- For all models, we use Adam optimizer Kingma & Ba (2015).

- For all models, we apply exponential learning rate decay of 0.99 and perform gradient clipping using max $l^2$-norm=1.

- For all GRUwE configurations, we use the same architecture for the hidden to observation function $F_{out}(\mathbf{g}_{t,\Delta T})$: Linear(hidden_dim, target_dim).

# 5 Hyperparameters for Next Observation Prediction Task on USHCN

We keep batch size fixed to 50, the number of training epochs to 100, learning rate decay to 0.99, with gradient clipping and perform a hyperparameter search for each model as follows:

### 5.0.1 mTAND

For mTAND, we perform a grid search over time embedding dimension = $\{32, 64, 128\}$, latent state dimension =$\{8, 10, 16, 20\}$, number of reference points=$\{32, 64, 128\}$ and learning rate =$\{0.1, 0.05, 0.01, 0.001\}$. Of which, latent state dimension=8, number of reference points=32 and learning rate of 0.01 performs the best on the validation data.

### 5.0.2 GRU-D

For GRU-D, we perform a search over latent state dimension = $\{8, 10, 16, 20\}$ and learning rate = $\{0.1, 0.05, 0.01, 0.001\}$. We find that configuration with latent state dimension=20 and learning rate=0.01 performs best on the validation set.

### 5.0.3 CRU

Fixed hyperparameters for CRU are: variance activation for encoder='square', decoder='exp', transition='relu' encoder variance activation='square', decoder variance activation='exp', number of basis matrices=20, and the same encoder and decoder network architecture as used in Schirmer et al. (2022). We perform a search on latent state dimension=$\{8, 10, 16, 20\}$ and learning rate = $\{0.1, 0.05, 0.01, 0.001\}$. We report that the latent dimension=10 and learning rate=0.05 performs the best on validation set.

### 5.0.4 f-CRU

Fixed hyperparameters for f-CRU include: variance activation for encoder='square', decoder='exp', transition='relu' encoder variance activation='square', decoder variance activation='exp', number of basis matrices=20, and the same encoder and decoder network architecture as used in Schirmer et al. (2022). We perform a search on latent state dimension = $\{8, 10, 16, 20\}$ and learning rate = $\{0.1, 0.05, 0.01, 0.001\}$. We report that the latent dimension=10 and learning rate=0.05 performs the best on the validation set.

### 5.0.5 Latent ODE

We use Latent ODE model with ODE-RNN encoder. We perform a grid search on latent state dimension=$\{8, 10, 16, 20\}$, recognition network dimension=$\{16, 32, 64\}$, number of GRU units=$\{16, 32, 64\}$, number of generation layers=$\{2, 3\}$, number of recognition layers=$\{2, 3\}$ and learning rate= $\{0.1, 0.05, 0.01, 0.001\}$. The configuration that performs the best on validation split with latent state dimension, recognition network dimension, number of GRU units set to 20; learning rate of 0.01 and generation and recognition network layers set to 3.

### 5.0.6 ODE-RNN

For ODE-RNN, we use GRU as the RNN model and use the adjoint solver method implemented in the library: `https://github.com/rtqichen/torchdiffeq` for solving the ODEs in a differentiable manner. For ODE-RNN, we search over latent state dimension=$\{8, 10, 16, 20\}$ and the learning rates=$\{0.1, 0.05, 0.01, 0.001\}$. We find that the configuration of latent state dimension=20 and learning rate = 0.01 works the best on the validation set.

### 5.0.7 ContiFormer

We use the ContiFormer implementation released by the authors `https://github.com/microsoft/SeqML/tree/main/ContiFormer` in our implementation. Note that since USHCN has highest average sequence length, and ContiFormer is memory intensive, we can only fit a batch size of 4 samples in our GPU memory. Keeping other parameters fixed, we vary the latent state dimension = $\{8, 10, 16, 20\}$ and learning rate = $\{0.1, 0.05, 0.01, 0.001\}$. Our experiments show that latent state dimension=16 and learning rate=0.001 achieves the best performance on validation set.

### 5.0.8 GRU-$\Delta_t$

For GRU-$\Delta_t$, we search over latent state dimensions of the GRU =$\{8, 10, 16, 20\}$ and learning rates=$\{0.1, 0.05, 0.01, 0.005, 0.001\}$. We find latent state dimension=16 and learning rate=0.005 to be the best performing configuration.

### 5.0.9 RKN-$\Delta_t$

We use the RKN-$\Delta_t$ implementation made available by the authors `https://github.com/ALRhub/rkn_share.git`. Our RKN-$\Delta_t$ implementation uses the same encoders and decoders architecture as the CRU model. Keeping other parameters fixed, we search over latent state dimensions $= \{8, 10, 16, 20\}$ and learning rates $= \{0.1, 0.05, 0.01, 0.005, 0.001\}$. Latent state dimension=20 and learning rate=0.001 results in the best performing model.

### 5.0.10 T-PatchGNN

We perform a grid search over learning rates $= \{0.1, 0.05, 0.01, 0.005, 0.001\}$, time and node embedding dimensions $= \{4, 8, 16\}$, number of patches=$\{2, 4\}$ (more number of patches results in GPU OOM issue), and latent state dimension $= \{4, 8, 10, 12, 16\}$, while fixing the number of heads in one transformer layer = number of transformer layers = 1. We find that the configuration with learning rate=0.001, time and node embedding dimension=8, number of patches=2, latent state dimension=16 results in the best validation MSE.

### 5.0.11 GraFITi

We perform a grid search over learning rates $= \{0.1, 0.05, 0.01, 0.005, 0.001\}$, latent state dimension $= \{4, 8, 16, 20, 32, 64\}$, number of layers $= \{1, 2, 4\}$ and number of attention heads $= \{1, 2, 4\}$. We report that the configuration with lr=0.01, latent state dimension=8, number of layer=2 and number of attention heads=1 results in the best validation MSE.

### 5.0.12 GRUwE

We perform a grid search over latent state dimensions=$\{8, 10, 16, 20\}$ (to be comparable to other baselines considered) with learning rates $\{0.1, 0.05, 0.01, 0.005, 0.001\}$. For USHCN, the best model configuration that maximizes validation prediction MSE uses learning rate=0.1 and latent state dimension=20.

### 5.0.13 HyperIMTS

We perform a grid search over learning rates $= \{0.1, 0.05, 0.01, 0.005, 0.001\}$, latent state dimension $= \{8, 16, 20, 32, 64, 128\}$, number of layers $= \{1, 2, 4\}$ and number of attention heads $= \{1, 2, 4\}$. We report that the configuration with lr=0.001, latent state dimension=32, number of layer=2 and number of attention heads=2 results in the best validation MSE.

## 6 Hyperparameters for Next Observation Prediction Task on Physionet

We keep the following hyperparameters constant across all methods: batch size=100, number of training epochs=100, learning rate decay=0.99 and gradient clipping enabled. Below are the model specific experiments we carried out.

### 6.0.1 mTAND

For mTAND, we perform a grid search over time embedding dimension $= \{32, 64, 128\}$, latent state dimension $=\{8, 10, 16, 20, 22, 24\}$, number of reference points=$\{32, 64, 128\}$ and learning rate $= \{0.1, 0.05, 0.01, 0.001\}$. Of which, time embedding dim=32, latent state dimension=22, number of reference points=64 and learning rate=0.01 performs the best on the validation data.

### 6.0.2 GRU-D

For GRU-D, we perform a search over latent state dimension $= \{8, 10, 16, 20, 22, 24, 32, 64\}$ and learning rate $= \{0.1, 0.05, 0.01, 0.001\}$. We find that configuration with latent state dimension=16 and learning rate=0.01 performs best on the validation set.

### 6.0.3 f-CRU

Fixed hyperparameters for f-CRU include: variance activation for encoder='square', decoder='exp', transition='relu' encoder variance activation='square', decoder variance activation='exp', number of basis matrices=20, and the same encoder and decoder network architecture as used in Schirmer et al. (2022). We perform a search on latent state dimension $= \{8, 10, 16, 20, 32, 64\}$ and learning rate $= \{0.1, 0.05, 0.01, 0.001\}$. We report that the latent dimension=16 and learning rate=0.001 performs the best on the validation set.

### 6.0.4 CRU

Fixed hyperparameters for CRU are: variance activation for encoder='square', decoder='exp', transition='relu' encoder variance activation='square', decoder variance activation='exp', number of basis matrices=20, and the same encoder and decoder network architecture as used in Schirmer et al. (2022). We perform a search on latent state dimension=$\{8, 10, 16, 20, 32, 64\}$ and learning rate = $\{0.1, 0.05, 0.01, 0.005, 0.001\}$. We report that the latent dimension=32 and learning rate=0.005 performs the best on validation set.

### 6.0.5 Latent ODE

We use Latent ODE model with ODE-RNN encoder. We perform a grid search on latent state dimension=$\{8, 10, 16, 20, 32, 64\}$, recognition network dimension=$\{16, 32, 64\}$, number of GRU units=$\{16, 32, 64\}$, number of generation layers=$\{2, 3\}$, number of recognition layers=$\{2, 3\}$ and learning rate= $\{0.1, 0.05, 0.01, 0.005, 0.001\}$. The configuration that performs the best on validation split with latent state dimension, recognition network dimension, number of GRU units set to 32; learning rate=0.005 and generation and recognition network layers set to 3.

### 6.0.6 ODE-RNN

For ODE-RNN, we search over latent state dimension=$\{8, 10, 16, 20, 32, 64\}$ and the learning rates=$\{0.1, 0.05, 0.01, 0.005, 0.001\}$. We find that the configuration of latent state dimension=16 and learning rate = 0.001 works the best on the validation set.

### 6.0.7 ContiFormer

For ContiFormer, we perform a search over the latent state dimension = $\{8, 16, 20, 32, 64\}$ and learning rate = $\{0.1, 0.05, 0.01, 0.001\}$. Our experiments show that latent state dimension=16 and learning rate=0.001 achieves the best performance on the validation set.

### 6.0.8 GRU-$\Delta_t$

For GRU-$\Delta_t$, we search over latent state dimensions of the GRU =$\{8, 10, 16, 20, 24, 32, 64\}$ and learning rates=$\{0.1, 0.05, 0.01, 0.005, 0.001\}$. We find latent state dimension=32 and learning rate=0.005 to be the best performing configuration.

### 6.0.9 RKN-$\Delta_t$

Our RKN-$\Delta_t$ implementation uses the same encoders and decoders architecture as the CRU model. Keeping other parameters fixed, we search over latent state dimensions = $\{8, 10, 16, 20, 24, 32, 64\}$ and learning rates = $\{0.1, 0.05, 0.01, 0.005, 0.001\}$. Latent state dimension=32 and learning rate=0.001 results in the best performing model.

### 6.0.10 T-PatchGNN

We perform a grid search over learning rates = $\{0.1, 0.05, 0.01, 0.005, 0.001\}$, time and node embedding dimensions = $\{4, 8, 16\}$, number of patches=$\{5, 10, 20\}$, and latent state dimension = $\{4, 8, 16, 32, 64\}$, while fixing the number of heads in one transformer layer = number of transformer layers = 1. We find that the configuration with learning rate=0.001, time and node embedding dimension=8, number of patches=10, latent state dimension=8 results in the best validation MSE.

### 6.0.11 GraFITi

We perform a grid search over learning rates = $\{0.1, 0.05, 0.01, 0.005, 0.001\}$, latent state dimension = $\{4, 8, 10, 16, 20, 32, 64\}$, number of layers = $\{1, 2, 4\}$ and number of attention heads = $\{1, 2, 4\}$. We report that the configuration with lr=0.005, latent state dimension=64, number of layer=2 and number of attention heads=1 results in the best validation MSE.

### 6.0.12 GRUwE

We perform a search over learning rate=$\{0.1, 0.025, 0.05, 0.01, 0.005\}$, the hidden state dimensions= $\{10, 16, 20, 24, 32, 64\}$ (to be comparable to other models). For Physionet, we use learning rate=0.025 and latent state dimension=64.

### 6.0.13 HyperIMTS

We perform a grid search over learning rates = $\{0.1, 0.05, 0.01, 0.005, 0.001\}$, latent state dimension = $\{8, 16, 20, 32, 64, 128\}$, number of layers = $\{1, 2, 4\}$ and number of attention heads = $\{1, 2, 4\}$. We report

that the configuration with lr=0.005, latent state dimension=16, number of layer=2 and number of attention heads=4 results in the best validation MSE.

# 7 Hyperparameters for Next Observation Prediction Task on MIMIC-III-Small

We keep the following hyperparameters constant across all methods: batch size=100 (we reduce it for models if we face OOM issue), number of training epochs=100, learning rate decay=0.99 and gradient clipping enabled. Below are the model specific experiments we carried out.

### 7.0.1 mTAND

For mTAND, we perform a grid search over time embedding dimension = $\{32, 64, 128\}$, latent state dimension $=\{8, 16, 32, 64, 128\}$, number of reference points$=\{8, 32, 64, 128\}$ and learning rate = $\{0.1, 0.05, 0.01, 0.001, 0.0001\}$. Of which, time embedding dim=32, latent state dimension=128, number of reference points=8 and learning rate=0.0001 performs the best on the validation data.

### 7.0.2 GRU-D

For GRU-D, we perform a search over latent state dimension = $\{8, 10, 16, 32, 64, 128\}$ and learning rate = $\{0.1, 0.05, 0.01, 0.001\}$. We find that configuration with latent state dimension=64 and learning rate=0.001 performs best on the validation set.

### 7.0.3 f-CRU

Fixed hyperparameters for f-CRU include: variance activation for encoder='square', decoder='exp', transition='relu' encoder variance activation='square', decoder variance activation='exp', number of basis matrices=20, and the same encoder and decoder network architecture as used in Schirmer et al. (2022). We perform a search on latent state dimension = $\{8, 10, 16, 20, 32, 64\}$ and learning rate = $\{0.1, 0.05, 0.01, 0.001, 0.0001\}$. We report that the latent dimension=64 and learning rate=0.0001 performs the best on the validation set.

### 7.0.4 CRU

Fixed hyperparameters for CRU are: variance activation for encoder='square', decoder='exp', transition='relu' encoder variance activation='square', decoder variance activation='exp', number of basis matrices=20, and the same encoder and decoder network architecture as used in Schirmer et al. (2022). We perform a search on latent state dimension=$\{8, 10, 16, 20, 32, 64\}$ and learning rate = $\{0.1, 0.05, 0.01, 0.005, 0.001\}$. We report that the latent dimension=32 and learning rate=0.005 performs the best on validation set.

### 7.0.5 Latent ODE

We use Latent ODE model with ODE-RNN encoder. We perform a grid search on latent state dimension=$\{8, 10, 16, 20, 32, 64\}$, recognition network dimension=$\{16, 32, 64\}$, number of GRU units=$\{16, 32, 64\}$, number of generation layers=$\{2, 3\}$, number of recognition layers=$\{2, 3\}$ and learning rate= $\{0.1, 0.05, 0.01, 0.005, 0.001\}$. The configuration that performs the best on validation split with latent state dimension, recognition network dimension, number of GRU units set to 32; learning rate=0.001 and generation and recognition network layers set to 3.

### 7.0.6 ODE-RNN

For ODE-RNN, we search over latent state dimension=$\{8, 10, 16, 20, 32, 64\}$ and the learning rates=$\{0.1, 0.05, 0.01, 0.005, 0.001, 0.0001\}$. We find that the configuration of latent state dimension=64 and learning rate = 0.0001 works the best on the validation set.

### 7.0.7 ContiFormer

For ContiFormer, we perform a search over the latent state dimension = $\{8, 16, 20, 32, 64\}$ and learning rate = $\{0.1, 0.05, 0.01, 0.001\}$. Our experiments show that latent state dimension=16 and learning rate=0.001 achieves the best performance on the validation set.

### 7.0.8 GRU-$\Delta_t$

For GRU-$\Delta_t$, we search over latent state dimensions of the GRU $=\{8, 10, 16, 20, 24, 32, 64\}$ and learning rates=$\{0.1, 0.05, 0.01, 0.005, 0.001\}$. We find latent state dimension=32 and learning rate=0.005 to be the best performing configuration.

### 7.0.9 RKN-$\Delta_t$

Our RKN-$\Delta_t$ implementation uses the same encoders and decoders architecture as the CRU model. Keeping other parameters fixed, we search over latent state dimensions = $\{8, 10, 16, 20, 24, 32, 64\}$ and learning rates = $\{0.1, 0.05, 0.01, 0.005, 0.001, 0.0001\}$. Latent state dimension=64 and learning rate=0.0001 results in the best performing model.

### 7.0.10 T-PatchGNN

We perform a grid search over learning rates = $\{0.1, 0.05, 0.01, 0.005, 0.001, 0.0001\}$, time and node embedding dimensions = $\{4, 8, 16\}$, number of patches=$\{5, 10, 20\}$, and latent state dimension = $\{4, 8, 16, 32, 64\}$, while fixing the number of heads in one transformer layer = number of transformer layers = 1. We find that the configuration with learning rate=0.0001, time and node embedding dimension=16, number of patches=10, latent state dimension=16 results in the best validation MSE.

### 7.0.11 GraFITi

We perform a grid search over learning rates = $\{0.1, 0.05, 0.01, 0.005, 0.001\}$, latent state dimension = $\{4, 8, 10, 16, 20, 32, 64\}$, number of layers = $\{1, 2, 4\}$ and number of attention heads = $\{1, 2, 4\}$. We report that the configuration with lr=0.005, latent state dimension=64, number of layer=2 and number of attention heads=1 results in the best validation MSE.

### 7.0.12 GRUwE

We perform a search over learning rate=$\{0.1, 0.025, 0.05, 0.01, 0.005\}$, the hidden state dimensions= $\{10, 16, 20, 24, 32, 64\}$ (to be comparable to other models). For MIMIC-III-Small, we use the configuration that results in the best validation MSE: learning rate=0.001 and latent state dimension=32.

### 7.0.13 HyperIMTS

We perform a grid search over learning rates = $\{0.1, 0.05, 0.01, 0.005, 0.001\}$, latent state dimension = $\{8, 16, 20, 32, 64, 128\}$, number of layers = $\{1, 2, 4\}$ and number of attention heads = $\{1, 2, 4\}$. We report that the configuration with lr=0.005, latent state dimension=32, number of layer=2 and number of attention heads=4 results in the best validation MSE.

## 8 Hyperparameters for Next Observation Prediction Task on MIMIC-III-Large

We keep the following hyperparameters constant across all methods: batch size=1 (to be able to handle the longer sequences in the MIMIC-III dataset within GPU memory constraints), number of training epochs=20, learning rate decay=0.99 and gradient clipping enabled. Below are the model specific experiments we carried out.

### 8.0.1 mTAND

Based on MIMIC-III experiments in Shukla & Marlin (2021), we keep the following hyperparameters fixed: time embedding dimension=128. We perform grid search on the hidden state dimension = $\{16, 32, 64\}$, encoder hidden dimension = $\{16, 32, 64\}$, the number of reference points = $\{64, 95\}$ and the learning rates=$\{0.01, 0.005, 0.001\}$. Note that for both hidden state dimensions as 64 and number of reference points as 95, we hit the memory limit on our GPU for batch size=1. Nonetheless, the resulting number of parameters (=174K) for this configuration is higher than that of our proposed model. Our validation results show that latent state dimension=64, number of reference points=95 and learning rate=0.001 performs the best across all combinations.

### 8.0.2 GRU-D

We perform grid search over hidden state dimension=$\{16, 32, 64\}$ and learning rates=$\{0.01, 0.005, 0.001\}$. We report that hidden state=32 and learning rate=0.001 performs the best on validation set and use it to report the final results.

### 8.0.3 Latent ODE

We use Latent ODE model with ODE-RNN encoder. We perform a grid search on latent state dimension=$\{16, 32, 64\}$, recognition network dimension=$\{16, 32, 64\}$, number of GRU units=$\{16, 32, 64\}$, number of generation layers=$\{2, 3\}$, number of recognition layers=$\{2, 3\}$ and learning rate=

$\{0.01, 0.005, 0.001\}$. The configuration that performs the best on validation split with latent state dimension, recognition network dimension, number of GRU units set to 32; learning rate of 0.001 and generation and recognition network layers set to 3. Other hyperparameters that were kept fixed are: batch size=1, learning rate decay=0.99 with gradient clipping.

### 8.0.4 f-CRU

We perform a grid search over the latent state dimensions= $\{16, 32, 64\}$ and learning rates=$\{0.01, 0.005, 0.001\}$. We set latent observation dimension as half the size of latent state dimension. Number of basis matrices = 20, and Gradient clipping enabled. Encoder consists of 3 $\times$( FullyConnected(50) + ReLU + Layer normalization) followed by linear output for latent observation and output; square activation for latent observation variance. Decoder consists of 3 $\times$ (FullyConnected(50) + ReLU + Layer normalization) followed by a linear output. Decoder output variance consists of (FullyConnected(50) + ReLU + Layer normalization) followed by linear output and square activation. Activation function for transition function is ReLU. After performing the grid search, the best configuration of hyperparameters are: latent state dimension=64, and learning rate=0.001.

### 8.0.5 CRU

Fixed hyperparameters for CRU are: variance activation for encoder='square', decoder='square', transition='relu', number of basis matrices=20, and the same encoder and decoder network architecture used in f-CRU (above). We perform a search on latent state dimension=$\{16, 32, 64\}$ and learning rate = $\{0.1, 0.01, 0.001, 0.0001\}$. We report that the latent dimension=32 and learning rate=0.0001 performs the best on validation set.

### 8.0.6 ODE-RNN

For ODE-RNN, we search over latent state dimension=$\{16, 32, 64\}$ and the learning rates=$\{0.1, 0.05, 0.01, 0.005, 0.001\}$. We find that the configuration of latent state dimension=32 and learning rate = 0.005 works the best on the validation set.

### 8.0.7 ContiFormer

For ContiFormer, we perform a search over the latent state dimension = $\{16, 32, 64\}$ and learning rate = $\{0.1, 0.05, 0.01, 0.001\}$. Our experiments show that latent state dimension=64 and learning rate=0.001 achieves the best performance on the validation set.

### 8.0.8 GRU-$\Delta_t$

For GRU-$\Delta_t$, we search over latent state dimensions of the GRU =$\{16, 32, 64\}$ and learning rates=$\{0.1, 0.05, 0.01, 0.005, 0.001\}$. We find latent state dimension=16 and learning rate=0.001 to be the best performing configuration.

### 8.0.9 RKN-$\Delta_t$

Our RKN-$\Delta_t$ implementation uses the same encoders and decoders architecture as the CRU model. Keeping other parameters fixed, we search over latent state dimensions = $\{16, 32, 64\}$ and learning rates = $\{0.001, 0.0005, 0.0001, 0.00005\}$. Note that we search over smaller values of the learning rate because, the model would not converge for higher ones (we get "NaN" during optimization for higher rate). Latent state dimension=32 and learning rate=0.00005 results in the best performing model.

### 8.0.10 T-PatchGNN

We perform a grid search over learning rates = $\{0.1, 0.05, 0.01, 0.005, 0.001\}$, time and node embedding dimensions = $\{4, 8, 16\}$, number of patches=$\{2, 4\}$ (more number of patches results in GPU OOM issue), and latent state dimension = $\{4, 8, 10, 12\}$, while fixing the number of heads in one transformer layer = number of transformer layers = 1. We find that the configuration with learning rate=0.001, time and node embedding dimension=20, number of patches=2, latent state dimension=12 results in the best validation MSE.

### 8.0.11 GraFITi

We perform a grid search over learning rates = $\{0.1, 0.05, 0.01, 0.005, 0.001\}$, latent state dimension = $\{16, 32, 64\}$, number of layers = $\{1\}$ (more number of layers on MIMIC-III causes OOM) and number of

attention heads = {1}. We report that the configuration with lr=0.005, latent state dimension=64, number of layer=1 and number of attention heads=1 results in the best validation MSE.

### 8.0.12 GRUwE

After searching over hidden state={10, 16, 20} and learning rates={0.1, 0.05, 0.01, 0.005, 0.001}. For the prediction tasks, we find the combination of 16 and 0.001 perform the best.

### 8.0.13 HyperIMTS

For HyperIMTS, we were unable to run experiments on the MIMIC-III Large dataset because even the smallest tested configuration (batch size = 1, latent dimension = 8, number of layers = 1, number of attention heads = 2) consistently triggered GPU memory errors during training. It is very likely due to long sequences in the MIMIC-III Large dataset as reported in Table 5.

## 9 Hyperparameters for Event Prediction task

We perform a grid search to tune the hyperparameters for the event prediction models: GRUwE, NHP, ATTNHP, THP and RMTPP. For RNN-based models (i.e., GRUwE, NHP, RMTPP), hidden size, learning rate and batch size are the relevant hyperparameters. For Attention-based models (i.e., ATTNHP, THP, SAHP), we optimize hidden size, learning rate, batch size, number of attention layers and embedding size. For all our experiments, we use Adam optimizer during training.

Table 6: Grid search configurations for hyperparameter tuning

| Model Type | Hyperparameter | Values |
|---|---|---|
| RNN-based (NHP, RMTPP, GRUwE) | hidden size
learning tate
batch size | {16, 32, 64}
{1e-1, 5e-2, 1e-2, 5e-3, 1e-3}
{32, 64, 128, 256} |
| Attention-based (ATTNHP, THP) | hidden Size
embedding size
num. layers
learning rate
batch size | {16, 32, 64}
{4, 8, 16}
{1, 2, 3}
{1e-1, 5e-2, 1e-2, 5e-3, 1e-3}
{32, 64, 128, 256} |

The grid search values for each parameter are reported in Table 6. For each dataset, we perform an independent hyperparameter search and train the model for up to 30 epochs with early stopping based on validation log-likelihood. We select the model configuration that achieves the highest validation log-likelihood.

## 10 Rank-based analysis for next event prediction.

We use the rank metrics as described in Table 7 to report the average rank statistic in the main results of our paper.

Table 7: Model performance comparison for the next event prediction on the Taxi, Retweet, StackOverflow, and Amazon datasets. Lower rank means better model for RMSE and ER metrics.

| Model | Taxi RMSE (Rank) | Taxi ER (Rank) | Retweet RMSE (Rank) | Retweet ER (Rank) | StackOverflow RMSE (Rank) | StackOverflow ER (Rank) | Amazon RMSE (Rank) | Amazon ER (Rank) |
|---|---|---|---|---|---|---|---|---|
| FullyNN | 0.373 (7) | N.A. | **21.92 (1)** | N.A. | 1.375 (6) | N.A. | 0.615 (5) | N.A. |
| NHP | **0.369 (1)** | 9.22 (3) | 22.32 (5) | **40.25 (1)** | 1.369 (2) | 55.78 (4) | 0.612 (2) | 68.30 (5) |
| A-NHP | 0.370 (4) | 11.42 (6) | 22.28 (3) | 41.05 (4) | 1.370 (4) | 55.51 (2) | 0.612 (2) | **65.65 (1)** |
| THP | **0.369 (1)** | 8.85 (2) | 22.32 (5) | **40.25 (1)** | **1.368 (1)** | 55.60 (3) | 0.612 (2) | 66.72 (2) |
| SAHP | 0.372 (6) | 9.75 (4) | 22.40 (7) | 41.60 (5) | 1.375 (6) | 56.10 (5) | 0.619 (6) | 67.70 (4) |
| RMTPP | 0.370 (4) | 9.86 (5) | 22.31 (4) | 44.10 (6) | 1.370 (4) | 57.50 (6) | 0.634 (7) | 73.66 (6) |
| GRUwE | **0.369 (1)** | **8.58 (1)** | 22.21 (2) | 40.81 (3) | 1.369 (2) | **55.34 (1)** | **0.611 (1)** | 67.24 (3) |

## 11   Computational Cost Analysis

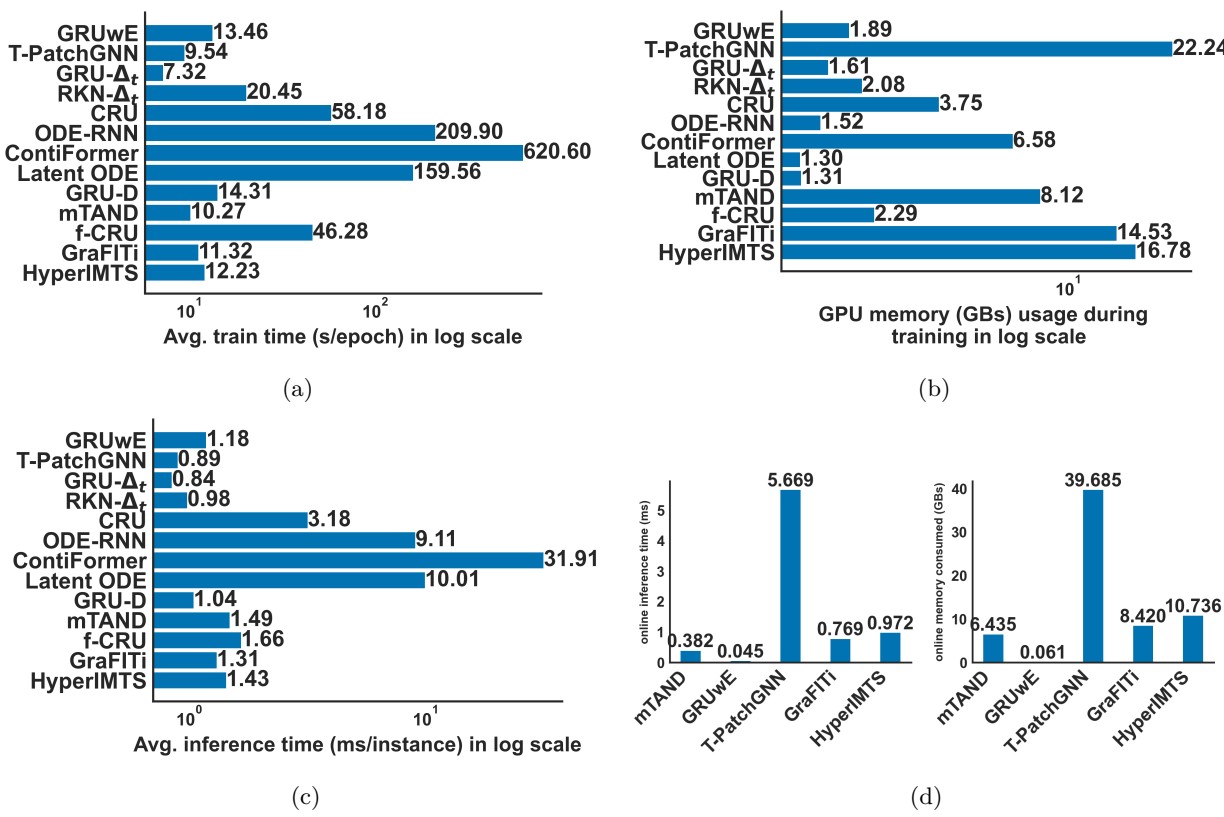

Figure 6: Computational cost analysis on the Physionet dataset. (**a**) Compares the average train time per epoch in seconds. (**b**) Peak GPU memory usage in GBs during training. (**c**) Average inference time in milliseconds per instance. (**d**) Comparison of the inference time (**left**) and memory consumption (**right**) in an online deployment.

We provide a comprehensive comparison of the computational complexity across all models investigated in our study in the Figure 6. We evaluate and contrast the training times, inference times, and peak GPU memory usage during training for each model when performing next observation prediction task on the Physionet dataset. The average train and inference times are computed by taking the average across multiple epochs and samples. Our analysis reveals a clear trend in terms of the train time. We note that methods (ContiFormer, ODE-RNN, Latent ODE) that rely on numerical solvers exhibit the longest training times, followed by models (CRU, f-CRU, RKN-$\Delta_t$) that assume linear dynamics. Recurrent models (GRUwE, GRU-D, GRU-$\Delta_t$) demonstrate moderate training times. Lastly, models that can be parallelized in the time dimension (mTAND, T-PatchGNN, GraFITi, HyperIMTS) are the fastest to train. However, it is crucial to note that the models with the fastest training times (mTAND, T-PatchGNN, GraFITi, HyperIMTS) come with a significantly higher memory requirement as illustrated in Figure 6b.

Having trained the time series model on retrospective EHR data, our primary goal is to deploy it as an early warning system to prevent adverse patient conditions in the ICU. This model operates in an inherently online context, characterized by a continuous stream of (near) real-time data. This data stream includes critical patient information such as vital signs and medication administration records, which the deployed model processes continuously. To simulate this online environment, we use the Physionet dataset, streaming observations from 100 randomly selected patients in an online manner. The models are evaluated based on total inference time and peak GPU memory usage during inference. In our analysis, we compare GRUwE with models that offer the best train times and are among the top performing models in next observation prediction task: mTAND, GraFITi and T-PatchGNN. We observe that since mTAND, GraFITi, T-PatchGNN and HyperIMTS do not have a Markovian state representation, it needs to buffer the past observations to make the inference. This results in higher inference cost both in terms of time and memory. Moreover, these

costs grows over time. In contrast, models that consists of Markov state representation such as GRUwE, are independent of the past observations thus, maintaining a constant, low time and memory requirement.

## 12   Computing Infrastructure

We used one server machine to deploy the experiments reported in the paper. This machine is equipped with 100GB memory, one NVIDIA L40S GPU, with Intel Xeon Platinum 8462Y+ @ 2.80 GHz processor and 16 CPU cores.

