# OpenReview forum: "Still Competitive: Revisiting Recurrent Models for Irregular Time Series Prediction"
_TMLR — Accepted by TMLR_

### Review · Reviewer_b7Hx · 2025-11-12

**Summary Of Contributions:**

This paper revisits recurrent neural network architectures for irregularly sampled time series, arguing that recent complex models may not justify their added complexity. The authors introduce GRU with exponential basis functions, which supports both regression and event prediction in continuous time through a Markov state representation with two reset mechanisms. The method demonstrates strong empirical performance while offering practical advantages in simplicity, ease of implementation, and fewer hyperparameters.
The abstract is clear and well-written, making the contribution easy to understand. The problem definition and analysis of exponential basis functions are technically sound, and results appear promising. However, the paper has a critical structural flaw: the related work is relegated to the appendix, leaving only a vague list of methods in the main text without adequate explanation of why this work matters or what issues exist with current approaches. This is particularly problematic given the paper's central claim that existing architectures are poorly understood.

**Audience:**

Yes

**Audience Explanation:**

This work explores a timely question in machine learning: Can simpler models achieve performance on par with their more complex counterparts? The answer holds significant practical value. By simplifying implementation and tuning without sacrificing performance, these methods could have broad appeal. Methodologically, the integration of exponential basis functions with GRU architecture and a dual reset mechanism introduces a concrete, actionable innovation that practitioners can readily adopt.

**Broader Impact Concerns:**

No significant concerns are identified. The paper addresses a methodological question without obvious negative ethical implications and does not involve sensitive applications.

**Claims And Evidence:**

Yes

**Claims Explanation:**

The technical content and experimental results appear sound, but the support for the paper's core motivation is missing.

**Requested Changes:**

- Substantially expand related work in the main paper.
This is the paper's most significant flaw and directly contradicts its thesis. Move key content from the appendix and replace the current list of methods without explanations with critical analysis explaining what problems exist with current architectures, why they are "poorly understood" with evidence, and how the proposed approach addresses these gaps. This section must make a compelling case for why revisiting RNNs matters.
- Provide evidence for the "poorly understood" claim. This is the central motivating claim and requires concrete examples or systematic literature review demonstrating that modern architectures lack clear understanding. If unsupportable, reframe the motivation.

- Add complete citations to Table 1.
Include missing recent baselines. Add HyperIMTS (ICML 2025) and other recent methods from 2024-2025. Provide explicit justification for any exclusions.
- Clarify Figure 1(a). Add clear y-axis label.

---

### Review · Reviewer_x2HC · 2025-11-15

**Summary Of Contributions:**

The paper introduces GRUwE, a novel yet intentionally simple recurrent model for irregularly sampled multivariate time series. The proposed architecture is evaluated through a comprehensive empirical study on two distinct tasks, demonstrating superior performance over existing methods. Additionally, the paper analyzes computational efficiency, showing that GRUwE is orders of magnitude faster and more memory-efficient in real-world online inference settings, compared to complex baselines that must re-process entire histories.

### Strengths:
- The model's simplicity and ease of implementation
- The computational efficiency in the online setting is a massive and well-proven strength, making it one of the few models tested that appears genuinely suitable for real-time deployment.
- The strong empirical performance
- The model is shown to be particularly effective on NMAR (Not Missing at Random) data, which is common in real-world scenarios like healthcare and a known challenge for many models.

### Weaknesses:
- The authors explicitly state that the model lacks a built-in mechanism for uncertainty quantification, which is a feature present in some of the more complex baselines.

**Audience:**

Yes

**Audience Explanation:**

It addresses the persistent challenge of modeling irregularly sampled multivariate time series , which is crucial in domains like healthcare. The paper presents a counter-intuitive finding that a simple, revisited RNN model can achieve competitive to superior performance compared to complex SOTA methods. The model is proven to have significantly lower computational overhead in real-time scenarios, a finding of high practical value to the ML community

**Claims And Evidence:**

Yes

**Claims Explanation:**

- Accuracy Claim: The claim of competitive to superior performance is supported by Table 1 and Table 2.

- Efficiency Claim: The claim of significantly lower computational overhead for online deployment is strongly supported by Figure 4a. It shows GRUwE is orders of magnitude faster (0.045 ms vs. 5.669 ms for T-PatchGNN) and uses dramatically less memory (0.061 GB vs. 39.685 GB) than competitors in an online setting.

- Simplicity Claim: The claim that GRUwE is easy to implement is visually confirmed by the pseudocode in Figure 4b, which shows the core mechanism in just four lines of code

**Requested Changes:**

- The paper's motivation hinges on the claim that the benefits of modern, complex architectures are 'poorly understood.' Could the authors please be more specific about which aspects of their benefits they believe are poorly understood?

- The learnable exponential decay is the core of the model's novelty. A small ablation study, even on one dataset, would be very insightful. For instance, how does the full GRUwE model compare to a version with a fixed, non-learnable exponential decay? This would more concretely demonstrate the value of learning the decay parameters.

- Specify the Number of Random Seeds: The paper states results are averaged over "multiple distinct random seeds". For clarity, reproducibility, and to assess statistical robustness, the authors should specify the exact number of seeds used for these experiments.

- Clarify Table Highlighting: The highlighting of top results in the tables (e.g., Table 1) appears inconsistent (how many). Please use a clear, uniform rule.

---

### Review · Reviewer_xkyY · 2025-11-15

**Summary Of Contributions:**

This paper challenges the trend of using increasingly complex (thus computationally expensive) architectures for modeling irregularly sampled time series. The paper further hypothesizes that classic architectures, if carefully modified, can just as good if not better. To support the claim, this paper modifies GRU and proposes GRUwE that maintains Markov state representation and uses a dual reset mechanism to handle the irregular time gaps. Empirical study shows the proposed approach is competitive comparing to strong baselines on various dataset. Runtime comparison also shows the much better efficiency of the proposed approach.

__Strengths__
- The proposed approach is technically sound.
- The empirical study provide evidence that supports the effectiveness and efficiency claim about the “carefully modified” “classic architecture”.
- The paper is easy to follow.

__Suggestions__
- Lack of Ablation Studies. The paper proposes an architecture that modifies the standard GRU with several new design choices, most notably the time-triggered exponential decay mechanism. However, the experiments only compare the final GRUwE model against other baselines. The paper does not provide an empirical breakdown (an ablation study) to demonstrate which of these new components are responsible for the performance gains. It's unclear if the entire design is necessary, or if one specific part is doing all the heavy lifting.
- The paper's motivation hinges on the strong claim that "the benefits of...modern architectures...remain poorly understood," but it doesn't provide citations or evidence to support this assertion. Concurrently, while the paper provides a theoretical analysis of GRUwE, it doesn't sufficiently connect this theory back to the empirical results to convincingly argue that GRUwE is, in contrast, "well understood.”

__Note__: as I am not familiar with this line of research (irregular Multivariate time series prediction), I cannot assess the novelty/significance of the proposed approach.

**Audience:**

Yes

**Audience Explanation:**

The research topic addressed by the paper is well within the scope of the machine learning community.

**Claims And Evidence:**

Yes

**Claims Explanation:**

The paper are in general convincing and evidences for supporting the claims are clear. The only concern is listed above in "suggestions" section.

**Requested Changes:**

Please see suggestions section above.

---

### Decision · Action_Editor_ajyn · 2025-12-30

**Recommendation:** Accept as is

**Audience:**

Yes

**Audience Explanation:**

TMLR readers working on time series, healthcare/sensor ML, and efficient sequence modeling will care because the paper provides a strong, deployment-friendly baseline that challenges the need for overly complex architectures. The main takeaway—well-designed recurrent models can remain highly competitive for irregular sampling—has immediate practical value and improves methodological clarity for the community.

**Claims And Evidence:**

Yes

**Claims Explanation:**

The paper's central claims—(i) a carefully designed GRU-style model can be competitive or better than recent complex irregular-TS architectures, and (ii) such a design yields major practical gains in online inference—are supported by empirical and analytic evidence in the revised manuscript. On the forecasting (next-observation) benchmarks, the results tables show GRUwE achieving top or near-top error across multiple real-world datasets, including strong performance on challenging high-dimensional clinical data (e.g., MIMIC-III-Large and MIMIC-III-Small) with multiple random seeds reported for robustness.